# Concealing Backdoors in Federated Learning by Trigger-Optimized Data Poisoning

## Abstract

Federated Learning (FL) is a decentralized machine learning method that enables participants to collaboratively train a model without sharing their private data. Despite its privacy and scalability benefits, FL is susceptible to backdoor attacks, where adversaries poison the local training data of a subset of clients using backdoor triggers, aiming to make the aggregated model produce malicious results when the same backdoor conditions are met by an inference-time input. Existing backdoor attacks in FL suffer from common deficiencies: fixed trigger patterns and reliance on the assistance of model poisoning. State-of-the-art defenses based on analyzing clients' model updates exhibit a good defense performance on these attacks because of the significant divergence between malicious and benign client model updates. To effectively conceal malicious model updates among benign ones, we propose DPOT, a backdoor attack strategy in FL that dynamically constructs backdoor objectives by optimizing a backdoor trigger, making backdoor data have minimal effect on model updates. We provide theoretical justifications for DPOT's attacking principle and display experimental results showing that DPOT, via only a *data*-poisoning attack, effectively undermines state-of-the-art defenses and outperforms existing backdoor attack techniques on various datasets.

## 1 Introduction

Federated Learning (FL) is a decentralized machine-learning approach that has gained widespread attention recently. Unlike traditional centralized model training, FL enables model updates to be computed locally on distributed data, offering enhanced data privacy, reduced communication overhead, and scalability for a large number of clients. In each round of FL, a central server distributes a global model to participating clients, each of whom independently trains the model on its local data, and its model updates are aggregated by the server for updating the global model. Despite its advantages, FL has been proven susceptible to backdoor attacks (Bagdasaryan et al., 2020). Backdoor attacks in FL involve adversaries manipulating the local models of a subset of clients to learn backdoor information from poisoned data with triggers, causing the global model, after aggregating these compromised local models, to produce adversary-desired results when the same trigger conditions are met. In this work, we term clients manipulated by adversaries during local training as *malicious clients*, and those unaffected as *benign clients*.

Existing backdoor attacks in FL present two common deficiencies. First, the patterns of backdoor triggers are pre-defined by the attacker and remain unchanged throughout the entire attack process (Bagdasaryan et al., 2020). Consequently, the optimization objective brought by backdoored data (backdoor objective) is static and incoherent with the optimization objective of main-task data (benign objective), resulting in distinct differences in model updates after training. These malicious clients' model updates are therefore easily canceled out by robust aggregations. Second, many approaches rely on model-poisoning techniques to enhance the effectiveness of backdoor attacks. Implementing model-poisoning attacks requires attackers to change the training procedures of a certain number of genuine clients to make their local training algorithms different from other clients. However, achieving this condition is challenging, as advanced defense mechanisms (Riege et al., 2024) have introduced Trusted Execution Environments (TEEs) to ensure the secure execution of client-side training, making it harder to maliciously modify the training procedure.

Existing defenses against backdoor attacks in FL rely on a hypothesis that backdoor attacks will always cause the updating direction of a model to deviate from its original benign objective, because the backdoor objectives defined by backdoored data cannot be achieved within the original direction (Fung et al., 2020; Cao et al., 2021). However, the capabilities of backdoor attacks are not limited to this hypothesis. To counter this hypothesis, adversaries can align the updating directions of a model with respect to backdoor and benign objectives by strategically adjusting the backdoor objective. Applying this idea to FL, if the injection of backdoored data has minimal effect on a client's model updates, then detecting this client as malicious becomes challenging for defenses based on analyzing clients' model updates.

In this work, we propose **D**ata **P**oisoning with **O**ptimized **T**rigger (DPOT), a backdoor attack on FL that dynamically constructs the backdoor objective to continuously minimize the divergence between clients' model updates in the backdoored states and the non-attacked states. We construct the backdoor objective by optimizing the backdoor trigger such that the current round's global model exhibits minimal loss on the backdoored data. When the global model becomes more optimized to the backdoored data, further training on this data will lead to smaller updates to the global model's current state. Therefore, when a malicious client's local dataset is partially poisoned by the optimized trigger while the rest remains benign, the model updates produced on them can be dominated by benign model updates within a limited number of local training epochs. We provide theoretical justification that trigger optimization can cause small differences in a client's model updates between the non-attacked and backdoored states (in Appendix C). Our experiments demonstrate that these small differences enable malicious model updates to bypass defenses and integrate into global models, resulting in backdoored global models.

Without any assistance of model-poisoning techniques, DPOT can be conducted simply by executing a normal training process on the poisoned local data. To ensure the stealthiness of the trigger, we constrain its $L_0$-norm by developing two algorithms to separately optimize the placement and values of trigger pixels. To the best of our knowledge, we are the first to propose algorithms for generating an optimized trigger with free shape and placement while specifying its exact size.

We evaluated DPOT on four image data sets (FashionMNIST, FEMNIST, CIFAR10, and Tiny ImageNet) and four model architectures including ResNet and VGGNet. We assessed the attack effectiveness of DPOT under a variety of defense conditions, testing it against eleven defense strategies that are based on analyzing clients' model updates along with one defense strategy that uses client-side adversarial training to recover the global model (Zhang et al., 2023). We compared DPOT attack with three state-of-the-art data-poisoning backdoor attacks that employ fixed-pattern triggers, distributed fixed-pattern triggers (Xie et al., 2020), and partially optimized triggers (Zhang et al., 2024), respectively. Using a small number of malicious clients (5% of the total), DPOT outperformed existing data-poisoning backdoor attacks in effectively undermining defenses without affecting the main-task performance of the FL system.

## 2 RELATED WORK

### 2.1 BACKDOOR ATTACKS IN FL

FL is very vulnerable to backdoor attacks. As training data are privately held by clients, the security of data is hard to track and protect. We discussed existing backdoor attacks in FL for image classification tasks based on their important properties (more details can be found in Appendix B.2).

**With vs. Without model poisoning.** Backdoor attacks in FL primarily rely on data poisoning, where attackers embed triggers in local training data and alter labels to train malicious models. Model poisoning (Fang et al., 2020) is often introduced to strengthen these attacks, by directly manipulating clients' model updates or training algorithms. However, it is challenged by the security provided by Trusted Execution Environments (TEEs), which authenticate and protect client-side training. In contrast, data poisoning is easier for attackers to conduct and harder to prevent, as clients would gather data from open, vulnerable sources. DPOT attack only relys on data poisoning.

**Static objective vs. Dynamic objective.** A static objective in backdoor attack represents a predefined and unchanging objective that is independent to the training system's status, such as associating certain input features or patterns with incorrect predictions. Having static objectives make malicious model updates easier to detect due to their inconsistency with main-task optimization. In

contrast, a backdoor attack that adjusts its objective based on the training system's status is referred to as having a dynamic objective. For example, Gong et al. (2022) and Fang & Chen (2023) optimized the trigger pattern based on a hypothesis that maximizing the activation of certain neurons in the backdoored local model can enhance the attack's persistence on the global model; however, these lack theoretical justification and proof-of-concept implementations. Zhang et al. (2024) optimized triggers specifically for a corner case where the global model is directly trained to unlearn the trigger, but the effectiveness of its triggers in more general FL training scenarios remains unaddressed. DPOT dynamically adjusts objectives to minimize the impact of backdoored data on model updates, and is provably effective in general FL training scenarios.

$L_2$-norm vs. $L_0$-norm bounded trigger. Designing effective backdoor triggers requires ensuring their stealthiness. $L_2$-norm bounds restrict the magnitude of changes *added* by the trigger, resulting in subtle perturbations spreading within a single input (Lyu et al., 2023). $L_0$-norm bounds restrict the number of components (e.g., pixels in an image) that can be *replaced* by the trigger. $L_2$-norm bounded triggers require the attacker to access and alter a figure's values before it is physically printed for use, and they are easily disrupted and filtered during data preprocessing. An $L_0$-norm bounded trigger, given its stable shape, consistent values, and compact size, are easier to apply. However, current works with optimized $L_0$-norm bounded triggers (Zhang et al., 2024; Fang & Chen, 2023) are still limited by fixing triggers' shape and placement, reducing their potential for more effective FL attacks. DPOT optimizes a trigger with an $L_0$-norm constraint, while allowing flexibility in its shape, placement, and value.

## 2.2 DEFENSES AGAINST BACKDOOR ATTACKS IN FL

In this work, we focus on defenses that adhere to the privacy-preserving principles of FL originally introduced by McMahan et al. (2017): clients' private data are kept local, and their model updates are not shared with any entities other than the server. For a discussion on additional defenses with varying privacy-preserving properties, please refer to the Appendix B.3.

In existing defenses, the server and clients are the two subjects commonly considered for implementing defense strategies. For benign clients as the defense subject, the global model of each round is the input they receive from the FL system. Zhang et al. (2023) proposed using trigger inversion on the global model and adversarial training on local models to mitigate the impact of the backdoor trigger. However, its effectiveness against continually evolving optimized triggers remains unaddressed. For server as the defense subject, clients' model updates are the input that the server receives from the FL system. Numerous studies proposed to defend against backdoor attacks by analyzing clients' model updates, which can be further classified into the two categories below.

**Excluding model updates with outlier values (in certain features).** Some existing works presume that a malicious client's model updates will exhibit significant differences from those of benign clients in values or certain features extracted from values. Nguyen et al. (2022) and Fung et al. (2020) exclude a client's model updates that have outlier cosine similarity in values to other clients' model updates. Sharma et al. (2023) and Ozdayi et al. (2021) reduce or penalize the contribution of model updates that show a certain degree of sign dissimilarity, either on a client-wise or element-wise basis. Kumari et al. (2023) and Fereidooni et al. (2024) assess the probabilistic distribution and frequency transformation of clients' model updates, and eliminate outliers in these features. Mozaffari et al. (2023) create a sparse space of model updates for clients to vote, and the server rejects outlier votes and aggregates the rest.

**Byzantine-robust aggregation.** Some existing works propose aggregating only the most trustworthy model updates to tolerate the presence of malicious clients. Yin et al. (2018) aggregate reliable model updates element-wise by taking median or trimmed mean, while Blanchard et al. (2017), Cao et al. (2022), and Pillutla et al. (2022) select and aggregate reliable model updates client-wise.

Analyzing clients' model updates can effectively defend against backdoor attacks that cause distinctions between malicious clients' and benign clients' model updates. However, when a backdoor attack can conceal malicious clients' model updates among benign ones, defenses based on this strategy will struggle (Bagdasaryan et al., 2020). In this work, we show that this goal can be achieved by dynamically changing the backdoor objectives defined on poison data, so that malicious clients' model updates are effectively manipulated.

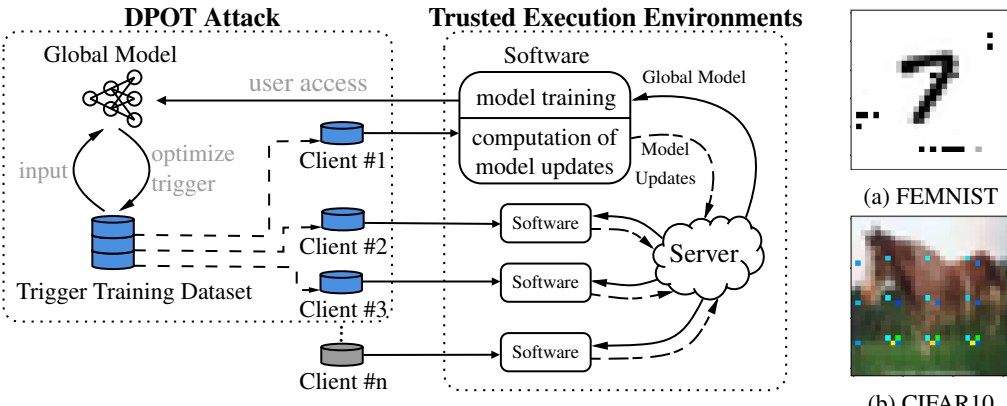

Figure 1: Overview of DPOT attack process on a FL system within Trusted Execution Environments (TEEs). In this figure, Client #1, #2, and #3 perform as the malicious clients while other clients (e.g. Client #n) are benign clients.

Figure 2: Data with DPOT triggers.

## 3 THREAT MODEL

**Attacker's capability and background knowledge:** As shown in Figure 1, we assume that each FL client—even a malicious one—is equipped with trustworthy training software that conducts correct model training on the client's local training data and transmits the model updates to the FL server. Aligning with the security settings in the state-of-the-art defense work (Riege et al., 2024), we assume that both the client training pipeline and the FL server, as well as the communication between them, faithfully serve FL's main task training and cannot be undetectably manipulated. These properties would be achievable by executing FL training within Trusted Execution Environments (TEEs) (Schneider et al., 2022; Riege et al., 2024), for example, by applying cryptographic protections to the updates (e.g., a digital signatures) to enable the FL server to authenticate the updates as coming from the TEEs.

The capability of malicious clients in our attack is limited to the manipulation of their local training data that are input to their training pipelines. In addition, in line with existing works (Lyu et al., 2023; Zhang et al., 2024; Fang & Chen, 2023; Gong et al., 2022), we do not assume the secrecy of the global model provided by the FL server, as it would typically need to be accessible outside TEEs for use in local inference tasks. As such, in each FL round, clients are granted white-box access to the global model. Originating from initially benign clients that have been compromised, these malicious clients possess some local training data for the FL main task as background knowledge.

**Attacker's goals:** The malicious clients aim to accomplish the following goals.

- **Effectiveness**. For classification tasks, *Attack Success Rate* ($ASR$) is the accuracy of a model in classifying data embedded with a backdoor trigger into a target label. The DPOT attack aims to make the post-aggregation global model misclassify data embedded with a trigger, optimized for the clients' most recently received global model, into a target label. Our effectiveness goal is for the global model to achieve an $ASR$ of over 50% in the final round and even maintain an average $ASR$ of over 50% across all rounds.
- **Stealthiness**. The stealthiness goal of a backdoor attack is to maintain the *Main-task Accuracy* ($MA$) of the global model at a normal level, ensuring the functionality of the global model on its main-task data.

## 4 DPOT DESIGN

### 4.1 BUILDING A TRIGGER TRAINING DATASET

At the beginning of the DPOT attack, we initially gather all available benign data from the malicious clients' local training datasets and assign a pre-defined target label $y_t$ to them. We refer to this new dataset, which associates benign data with the target label, as the trigger training dataset $D$.

**Algorithm 1** Computation for Trigger-pixel Placements

**Input:** $W_g, D, y_t, tri_{size}$
**Output:** $E_t$
  1: $\forall x \in D : y_x \leftarrow W_g(x)$.
  2: $\mathcal{L} \leftarrow \frac{1}{|D|} \sum_{x \in D} (y_x - y_t)^2$.
  3: $\forall x \in D : \delta_x \leftarrow \frac{\partial \mathcal{L}}{\partial x}$.
  4: $\delta \leftarrow abs(\sum_{x \in D} \delta_x)$.
  5: $\delta_f \leftarrow$ flatten $\delta$ into a one-dimensional array.
  6: $S \leftarrow \text{argsort}(\delta_f)$.
    *{ Store the sorted indices (descending sort)}*
  7: $E_t \leftarrow S[: tri_{size}]$.
    *{ Top $tri_{size}$ indices are trigger placements}*
  8: $E_t \leftarrow$ transform from one-dimensional indices to indices for $x \in D$.
  9: **return** $E_t$

**Algorithm 2** Optimization for Trigger-pixel Values

**Input:** $E_t, W_g, D, y_t, n_{iter}, \gamma$
**Output:** $V_t$
  1: **for** $iteration \leftarrow 1$ to $n_{iter}$ **do**
  2:     $D' \leftarrow D$.
  3:     **if** $iteration = 1$ **then**
  4:         $V_t \leftarrow \frac{1}{|D'|} \sum_{x \in D'} x$.
  5:     **else if** $iteration > 1$ **then**
  6:         $\forall x \in D' : x[E_t] \leftarrow V_t[E_t]$.
  7:     **end if**
  8:     $\forall x \in D' : y_x \leftarrow W_g(x)$.
  9:     $\mathcal{L} \leftarrow \frac{1}{|D'|} \sum_{x \in D'} (y_x - y_t)^2$.
 10:     $\forall x \in D' : \delta_x \leftarrow \frac{\partial \mathcal{L}}{\partial x}$.
 11:     $\delta \leftarrow \sum_{x \in D'} \delta_x$.
 12:     $V_t[E_t] \leftarrow (V_t - \gamma \cdot \delta)[E_t]$.
 13: **end for**
 14: **return** $V_t$

## 4.2 Optimizing Backdoor Trigger

**Formulating the optimization problem.** We generate a new backdoor trigger for each round's global model, with the optimization process operating independently across FL rounds. In this part, we introduce the trigger optimization algorithms within a single round of FL.

In the image classification context, consider the global model $W_g$ as input and all pixels within an image as the parameter space. Our approach aims to find a subset of parameters that have the most significant impact in producing the malicious output result (i.e., target label), and subsequently optimize the values of the parameters in this subset for the malicious objective (i.e., a high $ASR$). In the end, the pixels in this subset with their optimized values will serve as a backdoor trigger. This trigger will increase the likelihood that an image containing it will yield the malicious output when employing the same model $W_g$ for inference. The optimization objective to resolve the above problem can be written as formula (1).

$$\min_{\tau} \quad \frac{1}{|D|} \sum_{x \in D} Loss(W_g(x \odot \tau), y_t), \tag{1}$$

where $\tau$ represents the backdoor trigger composed of trigger-pixel placements $E_t$ and trigger-pixel values $V_t$. The objective is to minimize the difference between the target label $y_t$ and the output results of the global model $W_g$ when taking the backdoored images as input, which can be quantified by a loss function. The symbol $\odot$ represents an operator to embed the backdoor trigger $\tau$ into a clean image $x$, whose definition is further described in (2) of Appendix C. To enhance generalization performance of the trigger, we optimize it for all images in the trigger training dataset $D$.

**Compute trigger-pixel placements $E_t$.** In Algorithm 1, we select pixel locations that contain the largest absolute gradient values with respect to the backdoor objective (1) as the trigger-pixel placements. Algorithm 1 takes inputs including the global model $W_g$, the trigger training dataset $D$, the target label $y_t$, and a parameter $tri_{size}$ that specifies the trigger size. The trigger size $tri_{size}$ determines the number of pixel locations we will choose. The output of the Algorithm 1 is the trigger-pixel placement information denoted as $E_t$.

Starting from line 1 and line 2, we first calculate the loss of the global model $W_g$ in predicting clean images in dataset $D$ as the target label $y_t$, where we show Mean Square Error (MSE) as an example loss function. Next, we compute the gradient of the loss with respect to each pixel in each image and store the values of gradients in each image $x$ in $\delta_x$ (line 3). After summing up $\delta_x$ per pixel and take the absolute value of the results, we obtain an absolute gradient value matrix with the same shape as an individual image in dataset $D$ (line 4). To better describe how we sort elements in $\delta$ by their values, we first flatten $\delta$ into a one-dimensional array $\delta_f$ (line 5), and then sort elements in this array in descending order and store the sorted indices in an array $S$ (line 6). The top $tri_{size}$ number of indices are the trigger-pixel placements of interest, but before returning these indices, we transform them from indices for a one-dimensional array to indices for a matrix of an image's shape in dataset $D$ (line 7, line 8).

**Optimize trigger-pixel values** $V_t$**.** In Algorithm 2, we optimize the values of the trigger pixels defined in $E_t$ using a learning-based approach. Algorithm 2 requires the following inputs: the trigger-pixel placements $E_t$, the global model $W_g$, the trigger training dataset $D$, and the target label $y_t$. Additionally, it uses two training parameters: the number of training iterations $n_{iter}$ and the learning rate $\gamma$. The output produced by Algorithm 2 is the trigger-pixel value information denoted as $V_t$.

The first step of each iteration is making a copy dataset $D'$ of $D$ (line 2) so that the optimized trigger of each iteration can always be embedded into clean data. In the first iteration, we initialize the trigger-pixel value matrix $V_t$ by taking the mean value of all images in dataset $D'$ along each pixel location (line 4). Then, we calculate the loss of the global model $W_g$ in predicting images from $D'$ as the target label $y_t$ (line 8, 9). Next, we compute the gradients of the loss with respect to each pixel in each image in dataset $D'$ and store the values of gradients in each image x in $\delta_x$ (line 10). The gradient matrix $\delta$ is obtained by summing up $\delta_x$ along each pixel location (line 11) (but not need to take the absolute value as Algorithm 1). After that, we use the gradient descent technique with $\gamma$ as the learning rate to only update the values of pixels within the trigger-pixel placements $E_t$ (line 12) and assign those new values to the trigger value matrix $V_t$. For all iterations after the initial one, we consistently replace pixels within the trigger-pixel placements $E_t$ of each image with their corresponding values in the trigger value matrix $V_t$ (line 6). The steps of line 6 and line 12 ensure that the only variables influencing the loss result are the pixels specified by $E_t$.

### 4.3 Poisoning Malicious Clients' Training Data

The last step of our attack is to poison malicious clients' local training data using the optimized trigger $\tau = (E_t, V_t)$ and its target label $y_t$ by a certain data poison rate. The data poison rate can be specified on a scale from 0 to 1, while smaller data poison rate induces stealthier model updates, making them more difficult for defenses to detect and filter. In the following, we set the data poison rate to 0.5 for all experiments.

## 5 Experiments

### 5.1 Experimental Setup

**Datasets and global models:** We evaluated DPOT on four classification datasets with non-IID data distributions: Fashion MNIST, FEMNIST, CIFAR10, and Tiny ImageNet. Table 4 summarizes their basic information and models we used on each dataset.

**Comparisons:** As DPOT is exclusively a data-poisoning attack, we compared it with existing attacks where all the non-data-poisoning components were removed. We compared DPOT with three existing attacks as described below.

- **Fixed Trigger (FT)**. Following recent research on backdoor attacks on FL (Baruch et al., 2019; Xie et al., 2020; Cao et al., 2021; Bagdasaryan et al., 2020), we used a global pixel-pattern trigger with fixed features (values, shape, and placement) for all experiments in this attack category.
- **Distributed Fixed Trigger (DFT)**. DBA (Xie et al., 2020) proposed to slice a global pixel-pattern trigger into several parts and distributes them to malicious FL clients for data poisoning. The Attack Success Rate measures the effectiveness of this global trigger on the global model.
- **A3FL Trigger** A3FL (Zhang et al., 2024) proposed to adversarially optimize a trigger's value using a local model that continuously unlearns the optimized trigger information. The shape and placement of the A3FL trigger are fixed during optimization. Comparison results between DPOT and A3FL attacks are shown in Appendix E.1.

The visualization of various trigger types on different datasets are demonstrated in Appendix F.

**Defenses:** We evaluated backdoor attacks in FL systems employing different state-of-the-art defense strategies against backdoor attacks. We selected defenses that have open-sourced their proof-of-concept code to ensure accurate implementation of their proposed ideas. Here we present the evaluation results against ten defense strategies that rely solely on server-side execution; these defenses are described in Appendix D.1. In addition, evaluation results of defenses requiring client-side execution, Flip (Zhang et al., 2023) and FRL (Mozaffari et al., 2023), are given in Appendix E.2 and Appendix E.3 due to space limitations.

**Evaluation metrics:** We considered three metrics to evaluate the effectiveness and stealthiness of backdoor attacks when confronted with different defense strategies. To evaluate the effectiveness of the DPOT attack, we tested the post-aggregation global model using poisoned data with a trigger optimized for the clients' most recently received global model.

- **Final Attack Success Rate (Final $ASR$).** This metric quantifies the proportion of backdoored test images that were misclassified as the target label by the global model at the end of training. In order to reduce the testing error caused by noise on data or model so as to maintain the fairness of comparison, we tested $ASR$ on the global models of the last five rounds and took their mean value as the Final $ASR$.
- **Average Attack Success Rate (Avg $ASR$).** Since the attack cycle of DPOT spans just a single round, we introduced Avg $ASR$ to assess the average attack effectiveness across all rounds during the FL process. The implication of a high Avg $ASR$ of an attack is that this attack had consistently significant effectiveness during the whole FL process, ensuring a high Final $ASR$ no matter when the FL process ended.
- **Main-task Accuracy ($MA$).** We evaluate this metric by testing the accuracy of a global model on its clean main-task test dataset. A backdoor attack is seen to be stealthy if its victim model does not show a noticeable reduction in $MA$ compared to the benign model.

**FL training configurations:** Please see details of FL training settings in Appendix D.2.

**Attack configurations:** Table 5 shows the default settings of DPOT attack for experiments.

- **Trigger size.** The number of pixels that a backdoor trigger can alter is specified by the trigger size attribute. Selection of trigger sizes for various datasets are discussed in Appendix D.3.
- **Round.** We determine the number of training rounds for each dataset by measuring the convergence time on an FL pipeline using FedAvg as the aggregation rule. Convergence is considered achieved when the test accuracy on the main task stabilizes within a range of 0.5 percentage points over a period of five consecutive rounds of training.
- **Number of clients.** The number of clients varies across different datasets due to a balance between our available computational resources and the size of the datasets/models. All clients participate in the aggregation for each round of FL training.
- **MCR.** Malicious Client Ratio (MCR) is a parameter defining the proportion of compromised clients compared to the total number of clients in each-round aggregation. We consider $5\%$ as the default MCR (for FL systems having 50 clients, 2 of them are malicious clients).
- **Local data poison rate.** It indicates the proportion of data manipulated by a backdoor attack relative to the total data available on each malicious client.

## 5.2 EXPERIMENTAL RESULTS

### 5.2.1 REPRESENTATIVE RESULTS

In this section, we present the performance of DPOT attack against ten defense methods and compare our results with two widely-used data-poisoning attacks.

The effectiveness of an attack is measured using the $ASR$ metric, as shown in Figure 3. Results indicate that the DPOT attack consistently achieves a final $ASR$ exceeding 50% across all considered defense methods, regardless of the dataset's characteristics such as imgae size and number of images. Additionally, the DPOT attack also exhibits a considerable average $ASR$ in attacking different defenses, indicating its malicious effect on each-round global model. The stealthiness of an attack is assessed using the $MA$ metric, as indicated in Table 8. We established a baseline $MA$ for each defense method on every dataset by measuring the final $MA$ achieved in an attack-free FL training session employing the respective defense. Upon comparing the baseline $MA$ of various defenses to that of FedAvg, we observed that certain defenses, such as Multi-Krum on most of datasets and FLAME on Tiny ImageNet, failed to achieve similar convergence performance as FedAvg under the same training conditions. Defenses with deficient baseline $MA$ are less likely to be adopted in practice. The results presented in Table 8 indicate that the DPOT attack successfully maintains the $MA$ of victim global models within a $\pm 2$ percentage-point difference range compared to the corresponding baseline $MA$ values. In comparison to FT and DFT attacks, the DPOT attack demonstrates superior attack effectiveness in compromising existing defenses. As illustrated in Figure 3, the DPOT attack consistently demonstrates a higher Final $ASR$ compared to FT and DFT attacks and also achieves a significantly better Average $ASR$.

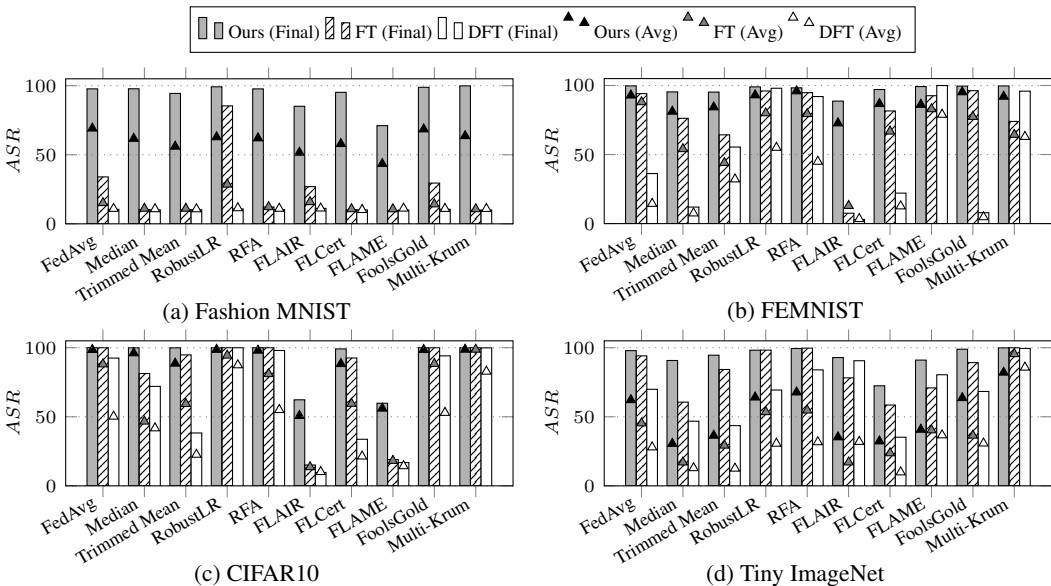

Figure 3: Representative results on four different datasets are provided. The attack settings correspond to the default settings outlined in Table 5.

### 5.2.2 ANALYSIS OF THE DPOT WORKING PRINCIPLES

In this section, we analyzed the attack effectiveness of each component of the DPOT attack's working principles and report evidence that it effectively conceals malicious clients' model updates, thereby getting them integrated into the global models through aggregation.

In the $i$-th round, DPOT generates a trigger $\tau^{(i)}$ by optimizing its shape, placement and values to make the global model of this round $W_g{}^{(i)}$ achieve a maximum $ASR$. However, what we were truly interested in is its $ASR$ on the global model after the $i$-th round aggregation, which is the next-round global model denoted as $W_g{}^{(i+1)}$. The attack effectiveness of the trigger $\tau^{(i)}$ on the global model $W_g{}^{(i+1)}$ stems from two factors:

1. **Trigger Optimization**: Trigger optimization using $W_g{}^{(i)}$ results in an improvement of the trigger's $ASR$ on $W_g{}^{(i+1)}$ due to the small difference between $W_g{}^{(i+1)}$ and $W_g{}^{(i)}$.

2. **Aggregation of Backdoored Model Updates**: Model updates that were trained on data partially poisoned by $\tau^{(i)}$ exhibit small differences from those were trained on data without poisoning. Therefore, they bypassed defenses and made $W_g{}^{(i+1)}$ incorporate backdoored model parameters.

In the following, we explain how we designed experiments to study the impact of each factor, and analyzed the experiment results.

**Experiment design:** To assess the attack effectiveness solely brought by Trigger Optimization, we eliminated any effects produced by data poisoning. Specifically, we set all clients in the FL system to be benign, ensuring that the next-round global model, denoted as $\widetilde{W}_g{}^{(i+1)}$, aggregated benign model updates only. In the meantime, we still collected data from a certain number of clients and optimized a trigger $\widetilde{\tau}^{(i)}$ for $\widetilde{W}_g{}^{(i)}$. Then, we tested $\widetilde{W}_g{}^{(i+1)}$ on a testing dataset in which all images are poisoned with the trigger $\widetilde{\tau}^{(i)}$ to obtain an $\widetilde{ASR}$. This $\widetilde{ASR}$ evaluates the attack effectiveness achieved by the current-round optimized trigger $\tau^{(i)}$ on the next-round global model $\widetilde{W}_g{}^{(i+1)}$, which does not contain any model updates learned from backdoor information.

To assess the attack effectiveness brought by Aggregation of Backdoored Model Updates, we introduced malicious clients into the FL system and therefore the global model, denoted as $\ddot{W}_g{}^{(i+1)}$, was allowed to aggregate model updates submitted by malicious clients. In this system, malicious clients partially poisoned their local training data (aligning with default settings in Table 5) using the trigger $\ddot{\tau}^{(i)}$ that was optimized for $\ddot{W}_g{}^{(i)}$, and then conducted their local training. We tested

the $\ddot{W}_g^{(i+1)}$ on the testing dataset that was also poisoned by $\ddot{\tau}^{(i)}$ to obtain an $A\ddot{S}R$. We evaluated the attack effectiveness of Aggregation of Backdoored Model Updates by measuring the increase in $ASR$ compared to the previous setting, calculated as $(A\ddot{S}R - \widetilde{ASR})$. This metric reveals how much the malicious clients' model updates influenced the global model $\ddot{W}_g^{(i+1)}$ to achieve a higher $ASR$ compared to $\widetilde{W}_g^{(i+1)}$.

Table 1: $ASR$ under different attacking conditions. $\widetilde{ASR}$ assesses the attack effectiveness of "Trigger Optimization" alone, while $A\ddot{S}R$ assesses the combined effectiveness of both "Trigger Optimization" and "Aggregation of Backdoored Model Updates".

| | | FedAvg | Median | Trimmed Mean | RobustLR | RFA | FLAIR | FLCert | FLAME | FoolsGold | Multi-Krum |
|---|---|---|---|---|---|---|---|---|---|---|---|
| **Fashion MNIST** | | | | | | | | | | | |
| Final | $\widetilde{ASR}$ | 58.8 | 57.9 | 31.6 | 70.2 | 78.0 | 42.2 | 49.6 | 38.0 | 54.2 | 60.6 |
| | $A\ddot{S}R$ | **97.7** | **97.8** | **94.4** | **99.2** | **97.7** | **85.3** | **95.2** | **71.1** | **98.9** | **99.9** |
| Avg | $\widetilde{ASR}$ | 45.1 | 38.2 | 29.7 | 47.2 | 46.4 | 36.2 | 39.7 | 26.2 | 50.3 | 45.4 |
| | $A\ddot{S}R$ | **69.1** | **61.7** | **56.0** | **62.8** | **62.0** | **50.1** | **57.9** | **43.4** | **68.5** | **63.6** |
| **FEMNIST** | | | | | | | | | | | |
| Final | $\widetilde{ASR}$ | 54.0 | 18.0 | 24.2 | 28.8 | 18.9 | 23.0 | 27.7 | 34.7 | 57.0 | 31.7 |
| | $A\ddot{S}R$ | **99.7** | **95.4** | **95.2** | **99.3** | **98.3** | **88.7** | **97.1** | **99.2** | **99.6** | **99.7** |
| Avg | $\widetilde{ASR}$ | 28.6 | 17.5 | 25.6 | 27.3 | 13.4 | 29.6 | 34.6 | 35.7 | 43.7 | 28.7 |
| | $A\ddot{S}R$ | **92.9** | **81.2** | **84.3** | **93.0** | **95.9** | **72.7** | **86.7** | **86.1** | **95.2** | **92.0** |
| **CIFAR10** | | | | | | | | | | | |
| Final | $\widetilde{ASR}$ | 55.6 | 56.6 | 55.6 | 60.1 | 57.4 | 54.1 | 48.7 | 28.1 | 35.5 | 49.7 |
| | $A\ddot{S}R$ | **100** | **100** | **100** | **100** | **100** | **62.3** | **99.2** | **59.8** | **100** | **100** |
| Avg | $\widetilde{ASR}$ | 50.9 | 48.7 | 40.9 | 47.3 | 46.1 | 45.9 | 46.7 | 51.0 | 35.6 | 36.1 |
| | $A\ddot{S}R$ | **98.5** | **96.1** | **88.6** | **98.6** | **97.8** | **50.7** | **88.3** | **56.1** | **98.5** | **98.7** |

**Experiment results:** Table 1 shows results of $\widetilde{ASR}$ and $A\ddot{S}R$ over 10 different defense methods. We used same settings as in Table 5 for testing $A\ddot{S}R$, and kept the size of trigger training dataset consistent when testing $\widetilde{ASR}$.

The results of $\widetilde{ASR}$ in Table 1 show that different defense methods resulted in very different $\widetilde{ASR}$ even for the same learning task of a dataset. The reason for the variance of $\widetilde{ASR}$ is the gap between $W_g^{(i)}$ and $\widetilde{W}_g^{(i+1)}$ were different when implementing different defense methods. According to recent studies (Lyu et al., 2023; Zhang et al., 2024), if the gap between consecutive rounds of global models in an FL system is smaller, Trigger Optimization will be more effective in its attack. The results of $A\ddot{S}R$ in Table 1 show that the presence of malicious clients' model updates consistently enhances $ASR$ compared to $\widetilde{ASR}$ across all defense methods on different datasets. We consider this enhancement as an evidence of the statement that the attack effectiveness of DPOT comes from both Trigger Optimization and Aggregation of Backdoored Model Updates, with the latter one playing a critical role in producing a high $A\ddot{S}R$.

A general hypothesis made by the state-of-the-art defenses against backdoor attacks in FL is that malicious clients' model updates have a distinct divergence from benign clients' model updates. However, as indicated by the results in Table 1, DPOT effectively conceals the model updates from malicious clients amidst those of benign clients, eluding detection and filtering by state-of-the-art defenses. Consequently, defenses formulated based on this broad hypothesis will inherently struggle to defend against DPOT attacks.

### 5.2.3 IMPACT OF MALICIOUS CLIENT RATIO (MCR)

In this section, we evaluated the impact of different Malicious Client Ratios (MCR) on the attacking performance of DPOT attack. We assumed that the number of malicious clients in the FL system should be kept small ($\leq 30\%$) for practical reasons. We varied the MCR across four different settings (0.05, 0.1, 0.2, and 0.3) while keeping other settings consistent with those in Table 5. We experimented over 10 different defenses on the learning tasks of the CIFAR10 datasets and compare DPOT's results with FT and DFT.

Tables 2 presents the evaluation results of attack effectiveness. DPOT exhibited a dominant advantage over FT and DFT when the MCR is small (0.05 and 0.1). However, this advantage diminished

Table 2: The effects of malicious client ratio on the effectiveness of different attacks (CIFAR10).

| MCR | Final $ASR$ | | | | | | | | | | | | Average $ASR$ | | | | | | | | | | | |
|---|---|---|---|---|---|---|---|---|---|---|---|---|---|---|---|---|---|---|---|---|---|---|---|---|
| | 0.05 | | | 0.1 | | | 0.2 | | | 0.3 | | | 0.05 | | | 0.1 | | | 0.2 | | | 0.3 | | |
| | Ours | FT | DFT | Ours | FT | DFT | Ours | FT | DFT | Ours | FT | DFT | Ours | FT | DFT | Ours | FT | DFT | Ours | FT | DFT | Ours | FT | DFT |
| FedAvg | **100** | **100** | 93 | **100** | **100** | 100 | **100** | **100** | 100 | **100** | **100** | 100 | **99** | 88 | 50 | **99** | 96 | 88 | **99** | 99 | 92 | 99 | **100** | 97 |
| Median | **100** | 81 | 72 | **100** | **100** | 97 | **100** | **100** | 100 | **100** | **100** | 100 | **96** | 47 | 42 | **97** | 79 | 63 | **99** | 97 | 82 | 99 | 98 | 93 |
| Trimmed Mean | **100** | 95 | 38 | **100** | **100** | 99 | **100** | **100** | 100 | **100** | **100** | 100 | **89** | 59 | 23 | **98** | 82 | 69 | **99** | 94 | 85 | 99 | 99 | 92 |
| RobustLR | **100** | **100** | 100 | **100** | **100** | 100 | **100** | **100** | 100 | **100** | **100** | 100 | **99** | 94 | 87 | **99** | 98 | 94 | **99** | 99 | 98 | 99 | 99 | 99 |
| RFA | **100** | **100** | 98 | **100** | **100** | 100 | **100** | **100** | 100 | **100** | **100** | 100 | **98** | 81 | 55 | **99** | 95 | 90 | **99** | 99 | 97 | 99 | 99 | 98 |
| FLAIR | **62** | 15 | 10 | **58** | 25 | 9 | **67** | 27 | 22 | **82** | 33 | 40 | **51** | 14 | 10 | **64** | 24 | 9 | **68** | 24 | 16 | **84** | 42 | 30 |
| FLCert | **99** | 93 | 34 | **100** | **100** | 95 | **100** | **100** | 100 | **100** | **100** | 100 | **88** | 60 | 21 | **98** | 87 | 60 | **98** | 94 | 83 | 99 | 99 | 91 |
| FLAME | **60** | 19 | 17 | **52** | 18 | 51 | **50** | 16 | 16 | **55** | 19 | 16 | **56** | 18 | 14 | **66** | 19 | 34 | **53** | 19 | 16 | **70** | 23 | 43 |
| FoolsGold | **100** | **100** | 94 | **100** | **100** | 100 | **100** | **100** | 100 | **100** | **100** | 100 | **98** | 88 | 53 | **99** | 97 | 87 | **99** | 99 | 95 | 99 | 99 | 98 |
| Multi-Krum | **100** | **100** | 100 | **100** | **100** | 100 | **100** | **100** | 100 | **100** | **100** | 100 | **99** | 99 | 83 | 99 | **100** | 98 | 98 | **100** | 99 | 99 | **100** | **100** |

Table 3: The effects of trigger size on the effectiveness of different attacks (CIFAR10).

| Trigger Size | Final $ASR$ | | | | | | | | | | | | Average $ASR$ | | | | | | | | | | | |
|---|---|---|---|---|---|---|---|---|---|---|---|---|---|---|---|---|---|---|---|---|---|---|---|---|
| | 9 | | | 25 | | | 49 | | | 100 | | | 9 | | | 25 | | | 49 | | | 100 | | |
| | Ours | FT | DFT | Ours | FT | DFT | Ours | FT | DFT | Ours | FT | DFT | Ours | FT | DFT | Ours | FT | DFT | Ours | FT | DFT | Ours | FT | DFT |
| FedAvg | **100** | 94 | 49 | **100** | **100** | 93 | **100** | **100** | 91 | **100** | **100** | 77 | **95** | 60 | 28 | **99** | 88 | 50 | **99** | 90 | 59 | 99 | 93 | 52 |
| Median | **97** | 23 | 12 | **100** | 81 | 72 | **100** | 95 | 25 | **100** | 99 | 46 | **66** | 21 | 12 | **96** | 47 | 42 | **98** | 66 | 17 | 99 | 82 | 29 |
| Trimmed Mean | **98** | 51 | 14 | **100** | 95 | 38 | **100** | 99 | 43 | **100** | 100 | 74 | **71** | 29 | 13 | **89** | 59 | 23 | **99** | 74 | 27 | 99 | 79 | 44 |
| RobustLR | **100** | **100** | 100 | **100** | **100** | 100 | **100** | **100** | 98 | **100** | **100** | 99 | **95** | 91 | 69 | **99** | 94 | 87 | **99** | 94 | 77 | 99 | 95 | 82 |
| RFA | **100** | **100** | 99 | **100** | **100** | 98 | **100** | **100** | 100 | **100** | **100** | 98 | **93** | 79 | 56 | **98** | 81 | 55 | **99** | 81 | 71 | 99 | 90 | 73 |
| FLAIR | **27** | 14 | 14 | **62** | 15 | 10 | **89** | 22 | 15 | **99** | 24 | 14 | **24** | 14 | 13 | **51** | 14 | 10 | **84** | 22 | 15 | 98 | 16 | 13 |
| FLCert | **99** | 38 | 14 | **99** | 93 | 34 | **100** | 88 | 51 | **100** | 100 | 49 | **78** | 26 | 13 | **88** | 60 | 21 | **99** | 59 | 23 | 99 | 78 | 33 |
| FLAME | **21** | 18 | 12 | **60** | 19 | 17 | **100** | 12 | 11 | **100** | 33 | 31 | **35** | 17 | 12 | **56** | 18 | 14 | **84** | 17 | 11 | 90 | 31 | 24 |
| FoolsGold | **100** | **100** | 43 | **100** | **100** | 94 | **100** | **100** | 98 | **100** | **100** | 81 | **93** | 72 | 23 | **98** | 88 | 53 | **99** | 94 | 69 | 99 | 94 | 55 |
| Multi-Krum | **100** | **100** | 15 | **100** | **100** | 100 | 99 | **100** | 100 | **100** | **100** | 100 | **99** | 99 | 11 | **99** | 99 | 83 | **99** | 99 | 95 | 99 | 99 | 97 |

with increasing MCR, indicating that when a sufficient number of malicious clients present in FL, even FT and DFT can achieve respectable $ASR$ against certain defense strategies. In most cases, the $ASR$ for all attacks continued to rise as the MCR increased, with the exception of FLAME. Results obtained with FLAME indicate that the number of malicious clients did not significantly impact its defense effectiveness. Table 9 presents the Main-task Accuracy results for each experiment considered in this section. All $MA$ results for different attacks remain similar to the baseline $MA$, indicating the correct implementation of each attack.

### 5.2.4 IMPACT OF TRIGGER SIZE

Trigger Size, determining how many pixels in an image we can alter, is an important parameter for DPOT attack. Larger trigger size generally results in a better optimization performance. However, it is essential to strike a balance because an excessively large trigger size will make a trigger obscure important details of images, making the trigger easier to perceive by humans. In this section, we assessed the impact of different trigger sizes on the performance of different attacks. We explored trigger sizes across four different settings (9, 25, 49, and 100) while maintaining other settings in accordance with those outlined in Table 5.

Tables 3 shows that DPOT maintained a significant advantage in $ASR$ over FT and DFT across various trigger sizes, ranging from small to large. According to the results, we found that FT and DFT did not benefit from larger trigger sizes in achieving higher $ASR$ when encountering with robust aggregations that have advanced defense effectiveness, such as FLAIR and FLAME. A possible explanation on that is when malicious model updates were trained on data poisoned with larger FT or DFT triggers, they exhibited greater divergence from benign model updates, making them more susceptible to detection and filtering by defense mechanisms. In contrast, DPOT demonstrated a continuous improvement in $ASR$ as the trigger size increased. Table 10 presents the Main-task Accuracy results for each experiment considered in this section. Results in it indicate all backdoor attacks achieved their stealthiness goals during attacking.

## 6 CONCLUSION

In this work, we proposed DPOT, a novel backdoor attack method relying solely on data poisoning in federated learning (FL). DPOT dynamically adjusts the backdoor objective to conceal malicious clients' model updates among benign ones, enabling global models to aggregate them even when protected by state-of-the-art defenses.

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

# Appendix

## Table of Contents

## A  ETHICS STATEMENT AND REPRODUCIBILITY STATEMENT

**Ethics Statement:**  Our paper presents a practical attack on federated learning, which can be executed with minimal technical skill by anyone who can participant into an FL. While this may seem risky, we believe the benefits of disclosing this attack outweigh potential harms. First, sharing the limitations of current defense strategies early prevents future misuse in security-critical applications, allowing organizations to address vulnerabilities before widespread deployment. Second, by publishing now, we provide the research community ample time to develop defenses, reducing long-term risks and maximizing the benefits of this work.

**Reproducibility Statement:**  To ensure the reproducibility of our results, we have provided detailed descriptions of our experimental setup, including model architectures, hyperparameters, datasets, and training procedures. All code used to implement our attack and run evaluations will be made available after the publication of this paper. Additionally, our code can be easily adapted to other FL research projects by simply integrating our algorithms into the data preparation process of FL

clients before the data is input into their training phase. Therefore, our work can be extensively used to evaluate future FL systems for security purposes.

## B   ADDITIONAL RELATED WORKS

### B.1   FEDERATED LEARNING (FL)

The Federated Learning (McMahan et al., 2017) (FL) training process involves four main steps: 1) **Model Distribution**: A central server distributes the most recent global model to the participating clients. 2) **Local Training**: Each client independently trains the global model on its local training dataset and obtains a local model. 3) **Model Updates**: Each client calculates the parameter-wise difference between its local model and the global model, referred to as model updates, and then sends them to the central server. 4) **Aggregation**: The central server aggregates clients' model updates to create a new global model. This entire process, consisting of step 1 to 4, constitutes a global round. The FL system repeats these steps for a certain number of rounds to obtain a final version of the global model.

### B.2   BACKDOOR ATTACKS IN FL

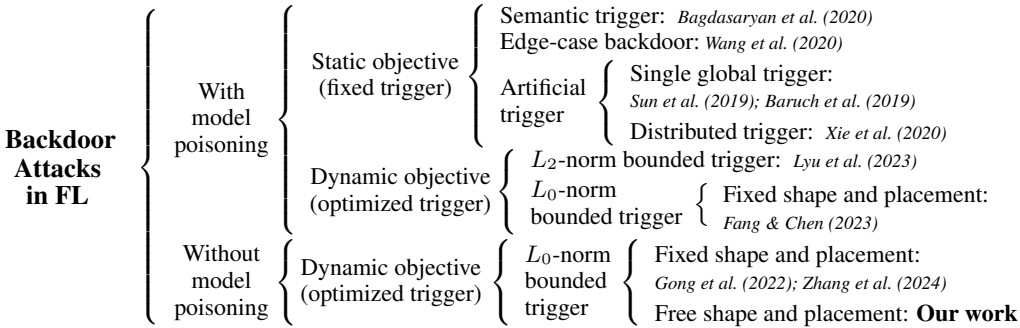

Figure 4: An overview of related works on backdoor attacks in FL.

FL is easily suffered from backdoor attacks. As training data are privately held by clients, the security of data is hard to track or protect. Adversaries can inject backdoors into the global model simply by compromising a few vulnerable client devices and poisoning their data with backdoor triggers. To date, many variations of backdoor attacks targeting FL have emerged, and we summarize those specific to image classification tasks in Figure 4.

**With model poisoning v.s. Without model poisoning**

The foundation of backdoor attacks in FL is through ***data poisoning*** - attackers embed backdoor triggers into the local training data of certain clients and change the ground-truth labels of the infected data to malicious labels. As a result, clients' local models trained on the poisoned data will be backdoored, and consequently, the global model that aggregates these backdoored models will also be backdoored.

A standalone data poisoning is found challenging to succeed when employing some types of triggers. Therefore, many works introduce model poisoning to assist backdoor attacks in FL. ***Model poisoning*** aims to either directly manipulate clients' model updates or indirectly achieve this by changing their local training algorithms. Three main approaches in model poisoning were widely adopted in existing attacks: 1) Scaling based (Bagdasaryan et al., 2020; Sun et al., 2019; Xie et al., 2020; Gong et al., 2022). Attackers amplify malicious model updates generated from backdoored models before clients send them to the server. These malicious updates can overpower the aggregation results, causing the global model to quickly incorporate backdoors. However, this approach is vulnerable to defenses that exclude outlier model updates from the aggregation. 2) Constraint based (Bagdasaryan et al., 2020; Lyu et al., 2023). Attackers change clients' local training algorithms by adding extra constraints to their loss functions, giving backdoored models specific characteristics, such as being less distinguishable from benign models. 3) Projection based (Zhang et al., 2022; Baruch et al., 2019; Wang et al., 2020; Fang & Chen, 2023). Attackers constrain backdoor implementation to

bounded model parameters: by clipping parameter values or using Projected Gradient Descent, backdoor models are $L_2$-norm bounded to a chosen model state; by selectively updating a subset of parameters, they are $L_0$-norm bounded to a chosen state.

Model poisoning requires attackers to modify certain clients' local training procedures. However, with the introduction of Trusted Execution Environments (TEEs) by state-of-the-art defense mechanisms (Riege et al., 2024), client-side execution for training can be authenticated and secure, thus increasing the difficulty of conducting model poisoning. In contrast, data poisoning is easier to conduct and harder to prevent since clients may collect their local data from open resources where attackers can also get access to and make modifications.

**Static objective v.s. Dynamic objective**

If a backdoor attack has a specified and unchanging objective that is independent to the training system's status, we refer to this as a ***static objective***. For instance, Semantic trigger as backdoor (Bagdasaryan et al., 2020) aims to associate certain features from input that is unrelated to the main training tasks with an attacker-chosen output, causing the model to make incorrect predictions on those inputs; Edge-case backdoor (Wang et al., 2020) selects data that share certain commonalities but are from the tail end of the input data distribution as the backdoored input, causing the model to mispredict them; Artificial trigger as backdoor (Sun et al., 2019; Zhang et al., 2022; Baruch et al., 2019; Xie et al., 2020) embeds a few pixels forming a specific artificial pattern into the input, leading the model to mispredict any input containing this pixel pattern. In FL, since the static objectives of backdoor attacks are inconsistent with the optimization objectives defined by the main-task data, malicious models will exhibit distinct differences in their model updates compared to benign models, making them easy to detect.

In contrast to a static objective, a backdoor attack that adjusts its objective based on the training system's status is referred to as having a ***dynamic objective***. By adjusting its objective, a backdoor attack is expected to achieve greater effectiveness. Several approaches have been proposed in recent attack studies to attempt to accomplish this. For example, Model-dependent attack (Gong et al., 2022) and F3BA (Fang & Chen, 2023) optimized the trigger pattern based on a hypothesis that maximizing the activation of certain neurons in the backdoored local model can enhance the attack's persistence on the global model, which is however lack of theoretical evidence and proof-of-concept codes; A3FL (Zhang et al., 2024) optimized triggers specifically for a corner case in FL training, where the global model is directly trained to unlearn the trigger, but the effectiveness of A3FL triggers in more general FL training scenarios remains unaddressed.

**$L_2$-norm bounded optimized trigger v.s. $L_0$-norm bounded optimized trigger**

A critical consideration in designing backdoor triggers is ensuring their stealthiness when applied to input data, resulting in a substantial disparity between human perception and the backdoored model's interpretation. Existing dynamic objective attacks achieve this by constraining the optimized triggers' $L_2$-norm or $L_0$-norm bounds.

An $L_2$-***norm bound*** on a trigger or perturbation means that the total magnitude of the changes introduced by the backdoor is limited. This makes the perturbation subtle, ensuring it doesn't drastically alter the input data. For example, CerP (Lyu et al., 2023) generates optimized perturbations of the same size as a data point for each round and adds them to clients' local data to induce their local models learn to misclassify the perturbed data to a specified target label.

An $L_0$-***norm bound*** restricts the number of components (e.g., pixels in an image) that can be altered by the trigger. This constraint ensures that the trigger is sparse, meaning it only affects a small portion of an input data. For example, optimized triggers in Model-dependent attack (Gong et al., 2022), F3BA (Fang & Chen, 2023), and A3FL (Zhang et al., 2024) all consist of a small number of pixels arranged in a square shape and are placed in a fixed corner location on the data to poison.

An $L_2$-norm bounded trigger is less practical for real-world data poisoning because it spreads changes across many pixels, requiring the attacker to access and alter a figure's values before it is physically printed for use. Additionally, these small perturbations are easily disrupted by data preprocessing techniques that filter out unnecessary noise. In contrast, an $L_0$-norm bounded trigger is easier to apply to data (e.g., a sticker on an image) due to its stable shape, consistent values, and compact size. However, existing works in optimizing $L_0$-norm bounded triggers are limited by fix-

ing their shapes and placements and only updating triggers' values, which fails to fully leverage the potential of optimized triggers for attacking FL.

**Clean-label attacks**

Clean-label attacks (Shafahi et al., 2018) involve manipulating input data with subtle perturbations while keeping labels unchanged. Although this assumption aligns with scenarios like Vertical Federated Learning (Liu et al., 2024) (VFL), where participants possess vertically partitioned data with labels owned by only one participant, our study does not consider VFL as our attack scenario. Furthermore, we focus on examining the effects of different backdoor triggers on hiding malicious model updates rather than their imperceptible characteristics. Therefore, discussions of clean-label attacks are beyond the scope of our work.

### B.3 Defenses with different privacy-preserving properties

Recent defense works have introduced several unconventional FL pipelines aimed at enhancing the security of FL against various types of attacks. These novel architectures provide different levels of privacy protection and often require additional techniques (e.g., Secured Multi-party Computation) to ensure privacy for FL clients. In light of these privacy considerations, we have chosen to focus our analysis on the conventional FL structure that was originally proposed in the concept of Federated Learning (McMahan et al., 2017). Although defenses built on newly proposed FL structures fall outside the scope of our main comparison, we offer a discussion of these related works in this section.

**Clients' private data were shared to the server:** Some approaches allow the server to have access to a small portion of main-task data shared by clients. To mitigate backdoor attacks, server-side defense strategies use this data to either independently train a model and use its updates as a reference for each round of aggregation (e.g., FLTrust (Cao et al., 2021)), or to validate clients' model updates and eliminate those with abnormal outputs (e.g., SSDT (Mo et al., 2024), SHERPA (Sandeepa et al., 2024)). However, both of these methods still rely on analyzing clients' model updates, making them vulnerable to backdoor attacks with dynamic objectives that conceal malicious updates. FedREdefense (Xie et al., 2024) detects and filters out artificial model updates by reconstructing distilled data shared by clients, but this approach is not effective against backdoor attacks where malicious clients genuinely train their models on poisoned local data rather than fabricating artificial updates.

**Clients' model updates were shared to each other:** Some approaches propose allowing clients to share their model updates with one another, rather than just with the server. CrowdGuard (Riege et al., 2024) and FLShield (Kabir et al., 2024) suggest that a subset of clients validate other clients' model updates using their own data, assuming that malicious clients' updates would produce abnormal outputs on benign data. However, this hypothesis fails when malicious clients' updates are indistinguishable from non-backdoored updates, a state that can be achieved through backdoor attack with optimized triggers. Fang et al. (2024) proposed a decentralized FL framework without a central server, where clients exchange model updates and apply Byzantine-robust aggregation using their own updates as a reference. Like other defenses that rely on analyzing clients' model updates, this approach is also vulnerable to backdoor attacks with optimized triggers.

## C Theoretical Analysis

In this section, we delve into the reasons behind DPOT's ability to successfully bypass state-of-the-art defenses, and analyze the improvements of an optimized trigger generated by our algorithms in assisting backdoor attacks, compared to a fixed trigger.

We use a linear regression model to explain the intuition of this work. Consider a regression problem to model the relationship between a data sample and its predicted values. We define $x \in \mathcal{D}^{1 \times n}$, where $\mathcal{D}$ is a convex subset of $\mathbb{R}$ as a data sample, and the vector $\hat{y} \in \mathbb{R}^{1 \times m}$ as its target values. The model $\beta \in \mathbb{R}^{n \times m}$ that makes $x\beta = \hat{y}$ is what we want to solve.

For any given data $x$, a backdoor attack is aiming to make the model $\beta$ fit both the benign data point $(x, \hat{y})$ and the corresponding malicious data point $(x_t, y_t)$. We use $y_t \in \mathbb{R}^{1 \times m}$ to represent the backdoor target values and specify that $y_t \neq \hat{y}$. $x_t \in \mathcal{D}^{1 \times n}$ is the data $x$ embedded with a trigger $\tau$ by the following operation.

$$x_t = x(I_n - E_t) + V_t E_t, \tag{2}$$

where $V_t \in \mathcal{D}^{1 \times n}$ is a vector storing the trigger $\tau$'s value information, and $E_t \in \{0, 1\}^{n \times n}$ is a matrix identifying the trigger $\tau$'s location information. $E_t$ specifies the location and shape of the trigger, defined as $E_t = diag(d_1, d_2, ..., d_n), d_i \in \{0, 1\}$, where $\sum_{i=1}^{n} d_i = k$. Here, $k$ defines the number of entries in the original $x$ that we intend to alter. The abbreviation $diag(\cdot)$ stands for a diagonal matrix whose diagonal values are specified by its arguments. $I_n$ is an $n \times n$ identity matrix.

**Definition C.1.** *(**Benign Loss and Benign Objective**) Let $x \in \mathcal{D}^{1 \times n}$ be a benign data sample, $\hat{y} \in \mathbb{R}^{1 \times m}$ be the predicted value of $x$, and $\beta \in \mathbb{R}^{n \times m}$ be the prediction model. The loss to evaluate the prediction accuracy of $\beta$ on the benign regression is*

$$L(x, \hat{y}) = \| x\beta - \hat{y} \|_2^2 . \tag{3}$$

*The optimization objective to solve for $\beta$ for this benign task is*

$$\min_{\beta} \quad L(x, \hat{y}). \tag{4}$$

**Definition C.2.** *(**Backdoor Loss and Backdoor Objective** ) Let $x_t$ be a backdoored data sample embedded with a trigger $\tau(V_t, E_t, y_t)$. Let $\beta \in \mathbb{R}^{n \times m}$ be the prediction model. The loss to evaluate the prediction accuracy of $\beta$ on the backdoor regression is*

$$L(x_t, y_t) = \| x_t\beta - y_t \|_2^2 . \tag{5}$$

*The optimization objective to solve for $\beta$ for the backdoor task that considers both benign data and backdoor data is*

$$\min_{\beta} \quad (1 - \alpha)L(x, \hat{y}) + \alpha L(x_t, y_t), 0 \le \alpha \le 1. \tag{6}$$

The FL global model learns backdoor information only when it integrates malicious clients' model updates that were trained for the backdoor objective. Due to the implementation of robust aggregation, backdoor attackers have to ensure their model updates have limited divergence from those trained on benign data to avoid being filtered out by defense techniques. We term this intention as the concealment objective.

To formulate the above problem, we use gradients of optimizing the benign objective ($G_{bn}$) and gradients of optimizing the backdoor objective ($G_{bd}$) with respect to a same model $\beta$ to represent model updates of a benign client and a malicious client respectively. We then use cosine similarity as a metric to evaluate the difference between $G_{bn}$ and $G_{bd}$, since it is a widely used metric in the state-of-the-art defenses Cao et al. (2021); Nguyen et al. (2022); Sharma et al. (2023); Fung et al. (2020) to filter malicious model updates.

$G_{bn}$ and $G_{bd}$ are computed by

$$G_{bn} = \frac{\partial L(x, \hat{y})}{\partial \beta}, \tag{7a}$$

$$G_{bd} = \frac{\partial((1 - \alpha)L(x, \hat{y}) + \alpha L(x_t, y_t))}{\partial \beta}. \tag{7b}$$

The concealment objective is

$$\max \quad CosSim(G_{bn}, G_{bd}). \tag{8}$$

The optimization objective used in DPOT attack is

$$\min_{V_t, E_t} \quad \| (x(I_n - E_t) + V_t E_t)\beta - y_t \|_2^2 . \tag{9}$$

**Proposition C.1.** *Given a model $\beta$ and a data sample $x$ with its benign predicted value $\hat{y}$ and a backdoor predicted value $y_t$, the optimization of objective (9) is a guarantee of the optimization of objective (8).*

*Proof.* The gradient of benign loss (3) with respect to $\beta$ is

$$g_{bn} = \frac{\partial L(x, \hat{y})}{\partial \beta} = 2x^T(x\beta - \hat{y}).$$

The gradient of backdoor loss (5) with respect to $\beta$ is

$$g_{bd} = \frac{\partial L(x_t, y_t)}{\partial \beta} = 2x_t^T (x_t \beta - y_t)$$

Gradients $G_{bn}$ and $G_{bd}$ defined by (7a) and (7b) can be written as

$$G_{bn} = g_{bn}.$$
$$G_{bd} = (1 - \alpha)g_{bn} + \alpha g_{bd}.$$

The cosine similarity between $G_{bn}$ and $G_{bd}$ is

$$CosSim(G_{bn}, G_{bd}) = \frac{g_{bn} \cdot ((1 - \alpha)g_{bn} + \alpha g_{bd})}{\mid g_{bn} \mid \cdot \mid (1 - \alpha)g_{bn} + \alpha g_{bd} \mid}$$
$$= \frac{g_{bn} \cdot (g_{bn} + \frac{\alpha}{1-\alpha}g_{bd})}{\mid g_{bn} \mid \cdot \mid g_{bn} + \frac{\alpha}{1-\alpha}g_{bd} \mid}$$

One sufficiency to maximize $CosSim(G_{bn}, G_{bd})$ is to minimize the distance between $g_{bn}$ and $g_{bn} + \frac{\alpha}{1-\alpha}g_{bd}$, which is

$$\Delta d = \mid g_{bn} - (g_{bn} + \frac{\alpha}{1 - \alpha}g_{bd}) \mid$$
$$= \frac{\alpha}{1 - \alpha} \mid g_{bd} \mid .$$

Since $\alpha$ is a constant, minimizing $\Delta d$ is equivalent to minimizing $\mid g_{bd} \mid$, which is bounded by

$$0 \leq \mid g_{bd} \mid = \mid 2x_t^T (x_t \beta - y_t) \mid \leq 2 \mid e^T \mid \cdot \mid x_t \beta - y_t \mid,$$

where $e^T \in \mathbb{R}^{1 \times n}$ consists of the largest edge of the domain of $x_t$, e.g. $\mathbf{1}^T$ if considering normalization.

Thus, the optimization objective is to decrease $\mid g_{bd} \mid$ by minimizing its upper bound.

$$\min \mid x_t \beta - y_t \mid,$$

which can be achieved by

$$\min_{V_t, E_t} \| (x(I_n - E_t) + V_t E_t)\beta - y_t \|_2^2 .$$

$\square$

Proposition C.1 offers a theoretical justification for DPOT's ability to prevent malicious clients' model updates from being detected by a commonly used metric considered in state-of-the-art defenses. In Proposition C.2 and Proposition C.3, we demonstrate that an optimized trigger ($\hat{\tau}$) generated by learning the parameters of a given model $\beta$ is more conducive to achieving the concealment objective compared to a trigger ($\tau_f$) with fixed value, shape, and location.

**Proposition C.2.** *For any fixed trigger $\tau_f(V_t, E_t, y_t)$ with specified trigger value $V_t$, trigger location $E_t$, and predicted value $y_t$, there exists an optimal backdoor trigger $\hat{\tau}(\hat{V}_t, E_t, y_t)$ that has the same $E_t$ and $y_t$ but optimizes its $V_t$ with respect to a model $\beta$, which can result in a smaller or equal backdoor loss on model $\beta$ compared to $\tau_f$.*

*Proof.* With a specified location $E_t$ and predicted value $y_t$, the optimization objective for minimizing backdoor loss is

$$f = \min_{V_t} \| (x(I_n - E_t) + V_t E_t)\beta - y_t \|_2^2$$

Since $V_t \in \mathcal{D}^{1 \times n}$ where $\mathcal{D}$ is a convex domain and $\frac{\partial^2 f}{\partial V_t^2} \succeq 0$ for any $V_t \in \mathcal{D}^{1 \times n}$, $f : \mathcal{D}^{1 \times n} \to \mathbb{R}$ is a convex function. Thus, there exists an optimal value $\hat{V}_t$ for the objective function $f$ in the domain $\mathcal{D}^{1 \times n}$. $\square$

**Proposition C.3.** *For any fixed trigger $\tau_f(V_t, E_t, y_t)$ with specified trigger value $V_t$, trigger location $E_t$, and predicted value $y_t$, there exists a backdoor trigger $\hat{\tau}(\hat{V}_t, \hat{E}_t, y_t)$ that has the same $y_t$, but optimizes the $V_t$ and $E_t$ with respect to a model $\beta$, which can result in a smaller or equal backdoor loss on model $\beta$ compared to $\tau_f$.*

*Proof.* Assume the value of data $x$ before embedding a trigger is $[x_1, x_2, ..., x_n]$. If an entry location in $x$ is able to reduce the backdoor loss of $\beta$ by optimizing its entry value more effectively than any individual entry location within $E_t$, we incorporate this location into $\hat{E}_t$. After constructing a $\hat{E}_t$, we optimize value of entries within $\hat{E}_t$ to obtain the optimized trigger $\hat{\tau}$. We are going to prove that constructing the trigger location $\hat{E}_t$ in this way results in the optimized trigger $\hat{\tau}$ always outperforming the fixed trigger $\tau_f$ in terms of backdoor loss.

We use $k$ to represent the number of trigger entries that have been embedded into $x$. Assume the trigger value $V_t$ is composed of $[v_1, v_2, ..., v_n]$.

When $k = 0$, the backdoor loss is

$$L(x, y_t)_{k=0} = \parallel x\beta - y_t \parallel_2^2 .$$

When $k = 1$, we calculate a location of interest $i$ by taking the largest absolute gradient of the $L(x, y_t)_{k=0}$ with respect to all entry locations in $x$,

$$i = arg \max | \frac{\partial L(x, y_t)_{k=0}}{\partial x_i} | .$$

If the entry location $i$ is inside of $E_t$, according to Proposition C.2, there exists an optimal entry value $\hat{v}_i$ resulting in a smaller or equal backdoor loss compared to $v_i$. In this case, we save $i$ as one of entry location in $\hat{E}_t$.

If the entry location $i$ is outside of $E_t$, we have the following observation:

For any entry location $j$ inside of $E_t$, we already know

$$| \frac{\partial L(x, y_t)_{k=0}}{\partial x_i} | \geq | \frac{\partial L(x, y_t)_{k=0}}{\partial x_j} | .$$

We use Gradient Descent optimization algorithm to decrease loss by updating the entry value of the selected location with a constant step size $\Delta v$. When the selected location is $i$, the updated loss $L(x^i, y_t)_{k=1}$ will be

$$L(x^{\{i\}}, y_t)_{k=1} = L(x, y_t)_{k=0} - \frac{\partial L(x, y_t)_{k=0}}{\partial x_i}\Delta v,$$

and when the selected location is $j$, it is

$$L(x^{\{j\}}, y_t)_{k=1} = L(x, y_t)_{k=0} - \frac{\partial L(x, y_t)_{k=0}}{\partial x_j}\Delta v.$$

It can be found that

$$L(x^{\{i\}}, y_t)_{k=1} \leq L(x^{\{j\}}, y_t)_{k=1}.$$

Therefore, $i$ is a better entry location in reducing backdoor loss compared to $j$ when we constrain the updating step size $\Delta v$ being static. After repeating the optimization step iteratively, if we finally find the optimal entry value $\hat{v}_i$ resulting in a smaller backdoor loss compared to $\hat{v}_j$, then it must also outperform the fixed value $v_j$ in $V_t$ according to Proposition C.2. If so, we save $i$ as one of entry location in $\hat{E}_t$. Otherwise, we save $j$ as one of location in $\hat{E}_t$.

By recursively operating the procedures across $k = 2, 3, ...$, we will finally construct a $\hat{E}_t$ in which every entry location is proved to contribute a better attack performance than entry locations defined in $E_t$.

$\square$

Table 4: Dataset description

| Dataset | #class | #img | img size | Model | #params |
|---|---|---|---|---|---|
| Fashion MNIST | 10 | 70k | $28 \times 28$ grayscale | 2 conv 3 fc | $\sim$1.5M |
| FEMNIST | 62 | 33k | $28 \times 28$ grayscale | 2 conv 2 fc | $\sim$6.6M |
| CIFAR10 | 10 | 60k | $32 \times 32$ color | ResNet18 | $\sim$11M |
| Tiny ImageNet | 200 | 100k | $64 \times 64$ color | VGG11 | $\sim$35M |

Table 5: Default settings

| | Trigger Size | Round | Number of Clients | MCR | Local Data Poison Rate |
|---|---|---|---|---|---|
| Fashion MNIST | 64 | 300 | 100 | | |
| FEMNIST | 25 | 200 | 100 | 0.05 | 0.5 |
| CIFAR10 | 25 | 150 | 50 | | |
| Tiny ImageNet | 64 | 100 | 50 | | |

# D EXPERIMENTAL SETUP

## D.1 DESCRIPTIONS OF DEFENSES

We implement our attack on FL systems integrated with 9 different defense strategies and provide a brief introduction for each of them:

**FedAvg (McMahan et al., 2017)**, a basic aggregation rule in FL, computes global model updates by averaging all clients' model updates. Despite its effectiveness on the main task, it is not robust enough to defend against backdoor attacks in the FL system.

**Median (Yin et al., 2018)**, a simple but robust alternative to FedAvg, constructs the global model updates by taking the median of the values of model updates across all clients

**Trimmed Mean (Yin et al., 2018)**, in our implementation, excludes the $40\%$ largest and $40\%$ smallest values of each parameter among all clients' model updates and takes the mean of the remaining $20\%$ as the global model updates.

**Multi-Krum (Blanchard et al., 2017)** identifys an honest client whose model updates have the smallest Euclidean distance to all other clients' model updates and takes this honest client's model updates as the global model updates. Despite its robustness to prevent the FL system from being compromised by a minor number of adversaries, Multi-Krum is not able to ensure the convergence performance of the FL system on its main task when the data distribution of clients is highly non-IID.

**RobustLR (Ozdayi et al., 2021)** adjusts the aggregation server's learning rate, per dimension and per round, based on the sign information of clients' updates.

**RFA (Pillutla et al., 2022)** computes a geometric median of clients' model updates and assigns weight factors to clients depending on their distance from the geometric median. Subsequently, it computes the weighted average of all clients' model updates to generate the global model updates.

**FLAIR (Sharma et al., 2023)** assigns different weight factors to clients according to the similarity of the coefficient signs between client model updates and global model updates of the previous round, and then takes the weighted average of all clients' model updates to form the global model updates. FLAIR requires the knowledge of exact number of malicious clients existing in the FL system.

**FLCert (Cao et al., 2022)** randomly clusters clients, calculates the median of model updates within each cluster, incorporates them into the previous round's global model, and derives the majority inference outcome from these cluster-updated global models as the final inference result for the entire FL system. In our implementation, we cluster clients into 5 groups, use FLCert inference outcome for testing the Attack Success Rate, and employ Median as the aggregation rule for updating the global model in each round.

**FLAME (Nguyen et al., 2022)** first clusters clients' model updates according to their cosine similarity to each other, and then aggregates the clipped model updates within the largest cluster as the global model updates.

**FoolsGold (Fung et al., 2020)** reduces aggregation weights of a set of clients whose model updates constantly exhibit high cosine similarity to each other.

### D.2 FL TRAINING CONFIGURATIONS

The FEMNIST dataset (Caldas et al., 2019) provides each client's local training data with a naturally non-IID guarantee. For Fashion MNIST, CIFAR10, and Tiny ImageNet datasets, we distributed training data to FL clients using the same method introduced by FLTrust (Cao et al., 2021), where we set the non-IID bias to be 0.5.

For all datasets training experiments, we used SGD optimization with CrossEntropy loss. In the experiments on Tiny ImageNet, we set the mini-batch size to 64, while for the other datasets, we set it to 256. Each FL client trained a global model for $n_{epoch} = 5$ local epochs with its local data in one global round.

For training Fashion MNIST and FEMNIST datasets, we used a static local learning rate ($lr$) of 0.01. For training the larger and more complicated datasets such as CIFAR10 and Tiny ImageNet, we applied the learning rate schedule technique following the instructions in the related machine learning works (He et al., 2016; Simonyan & Zisserman, 2015) to boost DNN models' performance.

### D.3 SELECTION OF TRIGGER SIZES

We defined trigger size for different datasets according to the following three criteria. First, a trigger of the defined trigger size should not be able to cover important details of any images and lead humans to misidentify the images from their original labels. To show that we follow this criteria, we demonstrated poisoned images from different datasets that are embedded with DPOT triggers, as shown in Figure 2 and 11. Second, on basis of the first criterion, we adjust trigger size to match the image size and the feature size of different datasets. Specifically, if a dataset contains images with high resolution (large image size), then a large trigger size is needed to effectively match it (Tiny ImageNet vs. CIFAR10). If images in a dataset contain large visual elements or patterns, then a large trigger size is needed to effectively match it (Fashion MNIST vs. FEMNIST). Third, we found that when using models with deep model architectures or having large number of parameters, a small trigger size is sufficient for conducting DPOT attack (CIFAR10 vs. Fashion MNIST).

### D.4 EXPERIMENT ENVIRONMENT AND CODE

We conducted all the experiments on a platform with multiple NVIDIA Quadro RTX 6000 Graphic Cards having 24 GB GPU memory in each chip and an Intel(R) Xeon(R) Gold 6230 CPU @2.10GHz having 384 GB CPU memory. We implemented all the algorithms using the PyTorch framework. We will open-source this project after its publication.

## E ADDITIONAL EXPERIMENTAL RESULTS

### E.1 COMPARISON TO A3FL TRIGGER

Table 6: Comparison results with A3FL attack on CIFAR10.

|  |  | FedAvg | Median | Trimmed Mean | Robust-LR | RFA | FLAIR | FLCert | FLAME | Fools-Gold | Multi-Krum |
|---|---|---|---|---|---|---|---|---|---|---|---|
| **Final** | A3FL | 48.9 | 32.9 | 35.0 | 46.2 | 24.7 | 13.2 | 39.0 | 13.7 | 46.9 | 33.4 |
| $ASR$ | Ours | **100** | **100** | **100** | **100** | **100** | **62.3** | **99.2** | **59.8** | **100** | **100** |
| **Average** | A3FL | 38.1 | 24.0 | 23.5 | 40.7 | 23.8 | 12.5 | 28.4 | 32.1 | 38.0 | 29.5 |
| $ASR$ | Ours | **98.5** | **96.1** | **88.6** | **98.6** | **97.8** | **50.7** | **88.3** | **56.0** | **98.5** | **98.7** |
| **MA** | A3FL | 70.6 | 69.1 | 69.9 | 71.2 | 70.2 | 70.7 | 69.9 | 70.1 | 70.8 | 62.8 |
|  | Ours | 70.7 | 69.1 | 70.4 | 70.1 | 70.7 | 70.6 | 70.0 | 70.3 | 71.0 | 63.0 |

In this section, we compared the performance of DPOT attack with the A3FL (Zhang et al., 2024) attack. We implemented the A3FL attack by faithfully replicating the attacker's actions as designed

by A3FL, with reference to their open-source project. We evaluated the effectiveness of A3FL attack against 10 defense strategies within our FL configurations and attack settings (refer to Table 5).

The results in Table 6 demonstrate that our attack achieved significantly higher $ASR$ values in both the final and average metrics compared to the A3FL attack. This suggests that the optimized triggers generated using our algorithms are more effective in compromising FL global models through data poisoning compared to those generated using A3FL's techniques. Additionally, we observed that the $ASR$ results of A3FL were even worse than those of FT and DFT (as shown in Figure 3) in our experiment settings. This implies that dynamically changing the backdoor objective may not enhance the effectiveness of backdoor attacks compared to maintaining a static backdoor objective if it can not align to the benign objective effectively.

## E.2 EVALUATION OF DPOT ATTACK AGAINST FLIP (ZHANG ET AL., 2023)

Flip (Zhang et al., 2023) is a client-side defense strategy where benign clients perform trigger inversion and adversarial training using their local data to recover the global model from backdoors. In this section, we evaluate the effectiveness of the DPOT attack against the Flip defense. We implemented the DPOT attack by modifying the data preparation approach in Flip's open-source project, replacing it with the method used in this work, and injecting our data-poisoning algorithms into a subset of clients. Additionally, as DPOT is a pure data-poisoning attack, we removed any additional steps in their project specified to malicious clients but not existed in benign clients' training, to ensure consistency between malicious clients and benign clients in FL training. We selected Fashion MNIST as the main-task dataset for our evaluation and directly adopted Flip's default experiment settings provided in their project - the total number of clients was 100 and 4% of them were malicious clients; the aggregation rule was set to FedAvg; the global model's parameters were initialized by a pre-trained state. The size of DPOT trigger was set to 64, consistent with our default attacking settings.

We compared the performance of the DPOT attack under two attack patterns provided by Flip's project: 1) **Single shot**: Each of the 4 malicious clients conducts a one-time attack at the beginning of training. 2) **Continuous**: All 4 malicious clients continuously execute the attack algorithms in every round during training.

Figure 5 shows the performance of the DPOT attack on an FL system using Flip as its defense, measured by the Attack Success Rate (ASR). In the single-shot attack pattern, DPOT maintains a stable ASR of around 15% across all training rounds, exceeding the random guess accuracy of 10% for the 10-class dataset. In the continuous attack pattern, DPOT achieves a significant ASR, peaking at 80.03% during training and stabilizing around 40%, which is higher than the single-shot pattern. These results indicate that Flip is vulnerable to optimized triggers with varying appearances across different rounds, because recovering from backdoors is an after-effect strategy which is unable to stop new and distinct backdoors from injecting into the model.

Figure 6 illustrates the global model's performance on the main task data when using Flip as a defense while under DPOT attack. We observed that employing Flip reduces the global model's main-task performance compared to not using it. In our baseline experiment on Fashion MNIST, with the same data distribution and aggregation rule (FedAvg), the model achieved an 86.7% MA. However, Flip's global model achieved only 82.8% MA at its best by the end, even with pre-trained model initialization. Additionally, under continuous attack by the DPOT trigger, the global model's MA further declined compared to the less frequent attack pattern. This raises concerns about Flip's ability to maintain stable and normal performance on the main task while effectively defending against attacks.

## E.3 EVALUATION OF DPOT ATTACK AGAINST FRL (MOZAFFARI ET AL., 2023)

FRL (Mozaffari et al., 2023) is a defense strategy where the server sparsifies the value space of model updates, allowing clients to vote on the most effective model updates based on their local data. The server then aggregates only the accepted votes while rejecting outliers to construct the global model. In this section, we evaluate the effectiveness of the DPOT attack against the FRL defense. Similar to the experiment on Flip, we implemented our attack on FRL's open-source project by injecting our data-poisoning algorithms into a portion of clients' execution and removing any inconsistent steps that distinguished malicious clients from benign ones during training. We used FRL's default

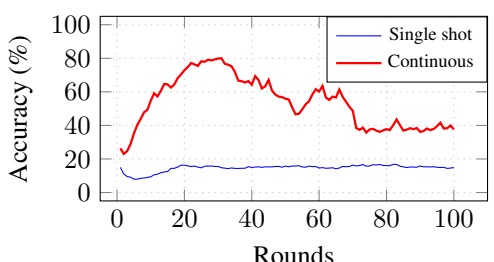
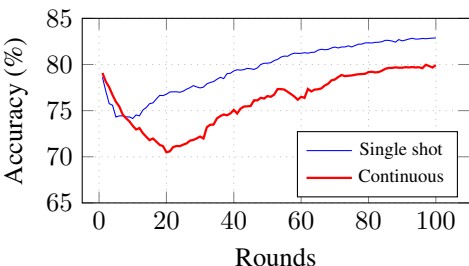

Figure 5: Global model's Attack Success Rate under DPOT attack when employed Flip as defense strategy. (Fashion MNIST)

Figure 6: Global model's Main-task Accuracy under DPOT attack when employed Flip as defense strategy. (Fashion MNIST)

settings, in which only 2% of clients were malicious, and tested our attack on the CIFAR10 dataset as the main training task.

Table 7 presents the performance results of the DPOT attack on an FL system employing FRL as the defense method. The ASR of DPOT (92.5%) is significantly higher than that of other backdoor attack approaches tested and discussed in FRL's paper. This indicates that FRL, which relies on analyzing clients' model updates, is vulnerable to our attack. The evaluation results also demonstrate that the DPOT attack is more advanced than backdoor attacks with static objectives when targeting the FRL defense strategy.

Table 7: Comparison results on CIFAR10.

| Attacks | ASR |
|---|---|
| Semantic backdoor attacks | 49.2 |
| Artificial backdoor attacks | 0 |
| Edge-Case backdoor attacks | 64.6 |
| **DPOT backdoor attacks** | **92.5** |

### E.4 EFFECTS OF THE SCALING-BASED MODEL POISONING TECHNIQUES ON ATTACKS

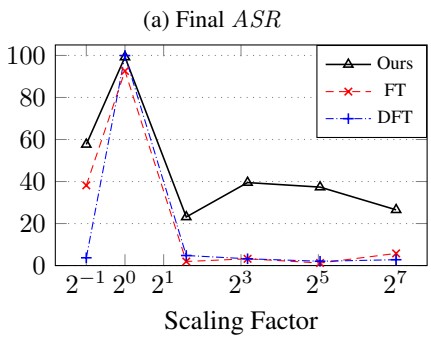
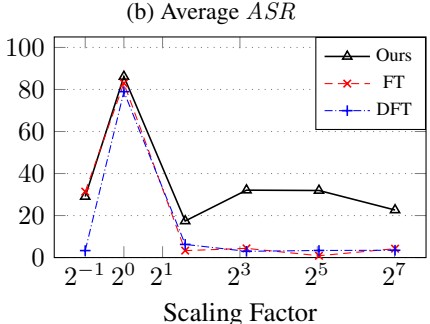

Figure 7: Comparison results of different attacks when employing the scaling-based model poisoning technique to undermine FLAME defense (implemented on the FEMNIST dataset).

In this section, we removed the TEEs assumption and conducted experiments to examine the effects of employing scaling-based model poisoning techniques on the attack performance of DPOT, FT, and DFT. By incorporating the model poisoning technique, our implementation of FT and DFT pipelines aligns more closely with the attack strategies introduced in state-of-the-art backdoor attacks on FL (Bagdasaryan et al., 2020; Xie et al., 2020).

Our experiments were designed within an FL system utilizing FLAME as its aggregation rule and FEMNIST dataset as its main training task. We adjusted the scaling factors, used to scale malicious

clients' model updates, to be 0.5, 1, 3, 9, 33, and 129 respectively. Figures 7a and 7b illustrate the results of Final $ASR$ and Avg $ASR$ of various attacks in response to different scaling factors.

We observed that when the scaling factor is 1, all DPOT, FT, and DFT pipelines exhibit comparable and high $ASR$ against FLAME defense. However, as the scaling factor increases, FLAME demonstrates robust defense performance, significantly reducing the $ASR$ of every attack pipeline. Despite this mitigation, DPOT shows greater resilience in attack effectiveness compared to FT and DFT. The optimized trigger generated by our algorithms retains intrinsic attack effects on the global model even without successful data-poisoning techniques. When the scaling factor is reduced to 0.5, malicious model updates are expected to be stealthier, yet their contributions to the aggregated global model are also mitigated, resulting in reduced $ASR$ for all attack pipelines compared to when the scaling factor is 1.

### E.5 MAIN-TASK ACCURACY RESULTS

Table 8 lists the Main-task Accuracy of each experiment in getting results in Figure 3. Table 8 demonstrates that for different datasets used as the main tasks, global models under various attacks maintained a comparable level of Main-task Accuracy to the baselines with no attacks ("None"), indicating that all types of backdoor attacks successfully achieved their stealthiness goals.

Table 9 lists the global model's Main-task Accuracy of each experiment in getting results in Table 2. Table 2 evaluates the impact of different malicious client ratios on the attack effectiveness of various attacks when using the CIFAR10 as the main-task dataset. Table 9 demonstrates that the performance of global models on Main-task data is not affected by changes in the malicious client ratio, indicating that the stealthiness goals of all backdoor attacks were achieved. "None" represents the baseline MA results with no attack present during FL training.

Table 10 lists the global model's Main-task Accuracy of each experiment in getting results in Table 3. Table 3 evaluates the impact of different trigger sizes on the attack effectiveness of various attacks when using the CIFAR10 as the main-task dataset. Table 10 demonstrates that the performance of global models on Main-task data is not affected by changes in the trigger sizes, indicating that the stealthiness goals of all backdoor attacks were achieved. "None" represents the baseline MA results with no attack present during FL training.

Table 8: The Main-task Accuracy (MA) of global models in getting representative results in Figure 3. "None" represents no attack existing in the FL training.

| MA | Tiny ImageNet | | | | Fashion MNIST | | | | FEMNIST | | | | CIFAR10 | | | |
|---|---|---|---|---|---|---|---|---|---|---|---|---|---|---|---|---|
| | None | Ours | FT | DFT | None | Ours | FT | DFT | None | Ours | FT | DFT | None | Ours | FT | DFT |
| FedAvg | 43.9 | 43.5 | 43.0 | 43.3 | 86.7 | 87.3 | 86.7 | 86.8 | 82.2 | 81.4 | 83.3 | 82.3 | 70.3 | 70.7 | 70.4 | 71.4 |
| Median | 40.6 | 40.2 | 40.6 | 38.6 | 86.0 | 85.8 | 86.6 | 86.3 | 80.4 | 81.5 | 79.8 | 79.9 | 70.2 | 69.1 | 69.8 | 69.7 |
| Trimmed Mean | 40.8 | 40.4 | 40.1 | 40.6 | 86.4 | 85.8 | 86.4 | 86.3 | 80.2 | 81.7 | 81.3 | 81.2 | 69.4 | 70.4 | 70.2 | 70.8 |
| RobustLR | 44.1 | 42.7 | 42.9 | 43.2 | 86.5 | 86.8 | 86.6 | 86.9 | 81.8 | 82.5 | 81.9 | 82.6 | 70.4 | 70.1 | 70.3 | 70.5 |
| RFA | 43.6 | 43.0 | 43.0 | 43.0 | 86.4 | 86.0 | 87.1 | 87.1 | 83.0 | 80.7 | 81.0 | 80.8 | 70.4 | 70.7 | 70.3 | 70.8 |
| FLAIR | 43.6 | 42.6 | 41.8 | 42.1 | 86.1 | 84.9 | 85.2 | 84.4 | 81.5 | 80.7 | 80.6 | 79.7 | 70.3 | 70.6 | 71.0 | 70.4 |
| FLCert | 40.3 | 40.2 | 39.7 | 39.7 | 86.2 | 85.9 | 86.0 | 86.8 | 81.3 | 80.9 | 81.5 | 81.0 | 69.6 | 70.0 | 69.8 | 70.4 |
| FLAME | 29.9 | 28.7 | 29.2 | 28.9 | 86.4 | 86.4 | 86.4 | 86.7 | 81.8 | 80.2 | 80.7 | 81.0 | 70.1 | 70.3 | 70.9 | 70.9 |
| FoolsGold | 43.1 | 43.2 | 43.5 | 43.2 | 86.6 | 87.1 | 86.8 | 87.3 | 83.4 | 82.7 | 83.0 | 81.8 | 70.4 | 71.0 | 71.2 | 71.7 |
| Multi-Krum | 30.7 | 27.7 | 27.7 | 26.4 | 86.2 | 85.9 | 86.0 | 87.0 | 79.9 | 80.4 | 79.6 | 80.2 | 61.4 | 63.0 | 63.2 | 60.8 |

Table 9: The Main-task Accuracy (MA) of global models under different attacks at varying malicious client ratios. (CIFAR10).

| MCR | | 0.05 | | | 0.1 | | | 0.2 | | | 0.3 | | |
|---|---|---|---|---|---|---|---|---|---|---|---|---|---|
| | None | Ours | FT | DFT | Ours | FT | DFT | Ours | FT | DFT | Ours | FT | DFT |
| FedAvg | 70.3 | 70.66 | 70.37 | 71.37 | 70.03 | 71.04 | 70.13 | 69.9 | 70.39 | 71.18 | 70.25 | 70.69 | 70.24 |
| Median | 70.21 | 69.06 | 69.76 | 69.71 | 69.32 | 69.17 | 70.12 | 68.23 | 69.05 | 68.87 | 68.49 | 68.47 | 67.82 |
| Trimmed Mean | 69.43 | 70.42 | 70.24 | 70.84 | 69.9 | 69.17 | 69.78 | 69.33 | 69.19 | 69.8 | 69.23 | 68.83 | 68.02 |
| RobustLR | 70.35 | 70.10 | 70.35 | 70.48 | 70.58 | 70.42 | 69.90 | 70.31 | 70.56 | 70.43 | 70.05 | 69.11 | 69.22 |
| RFA | 70.42 | 70.69 | 70.27 | 70.77 | 70.35 | 70.44 | 70.16 | 70.72 | 70.33 | 69.56 | 70.09 | 69.72 | 69.37 |
| FLAIR | 70.25 | 70.62 | 71.04 | 70.42 | 69.80 | 71.45 | 70.89 | 71.85 | 71.20 | 71.16 | 71.26 | 69.74 | 70.99 |
| FLCert | 69.6 | 69.95 | 69.76 | 70.42 | 69.44 | 69.44 | 69.45 | 69.28 | 69.25 | 69.73 | 68.54 | 69.06 | 68.24 |
| FLAME | 70.14 | 70.28 | 70.93 | 70.85 | 69.62 | 70.87 | 71.01 | 70.71 | 70.4 | 70.58 | 69.19 | 71.45 | 70.52 |
| FoolsGold | 70.42 | 71.02 | 71.19 | 71.68 | 70.71 | 71.32 | 71.27 | 70.45 | 70.38 | 70.82 | 70.12 | 69.97 | 69.97 |
| Multi-Krum | 61.38 | 62.98 | 63.16 | 60.80 | 61.44 | 62.89 | 62.09 | 59.38 | 61.26 | 63.70 | 60.28 | 64.02 | 62.96 |

Table 10: The Main-task Accuracy (MA) of global models under different attacks with varying trigger sizes. (CIFAR10).

| Trigger Size | | 9 | | | 25 | | | 49 | | | 100 | | |
|---|---|---|---|---|---|---|---|---|---|---|---|---|---|
| | None | Ours | FT | DFT | Ours | FT | DFT | Ours | FT | DFT | Ours | FT | DFT |
| FedAvg | 70.3 | 70.88 | 70.72 | 71.25 | 70.66 | 70.37 | 71.37 | 70.77 | 71.35 | 70.94 | 69.92 | 70.71 | 71.15 |
| Median | 70.21 | 68.31 | 70.04 | 68.69 | 69.06 | 69.76 | 69.71 | 69.95 | 70.54 | 70.56 | 69.88 | 70.30 | 70.86 |
| Trimmed Mean | 69.43 | 69.75 | 70.13 | 70.19 | 70.42 | 70.24 | 70.84 | 69.42 | 70.17 | 69.79 | 69.67 | 70.26 | 70.68 |
| RobustLR | 70.35 | 70.48 | 70.95 | 69.48 | 70.10 | 70.35 | 70.48 | 70.79 | 70.08 | 70.27 | 70.39 | 69.73 | 69.86 |
| RFA | 70.42 | 70.45 | 70.16 | 71.00 | 70.69 | 70.27 | 70.77 | 70.56 | 70.19 | 70.62 | 70.52 | 69.22 | 70.77 |
| FLAIR | 70.25 | 70.79 | 70.67 | 70.58 | 70.62 | 71.04 | 70.42 | 70.84 | 69.96 | 71.03 | 71.17 | 70.65 | 70.28 |
| FLCert | 69.6 | 69.88 | 69.64 | 69.87 | 69.95 | 69.76 | 70.42 | 67.77 | 69.83 | 70.08 | 68.81 | 70.81 | 70.41 |
| FLAME | 70.14 | 70.07 | 71.24 | 70.19 | 70.28 | 70.93 | 70.85 | 69.87 | 71.20 | 70.68 | 67.24 | 71.06 | 70.75 |
| FoolsGold | 70.42 | 70.4 | 72.1 | 70.09 | 71.02 | 71.19 | 71.68 | 70.66 | 70.75 | 71.38 | 69.84 | 71.06 | 71.64 |
| Multi-Krum | 61.38 | 62.86 | 64.65 | 58.90 | 62.98 | 63.16 | 60.80 | 58.23 | 60.16 | 64.04 | 63.03 | 61.64 | 63.33 |

# F    VISUALIZATION OF TRIGGERS

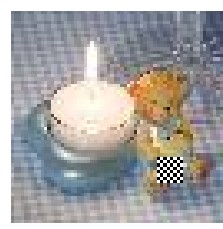 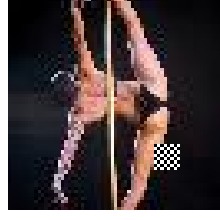 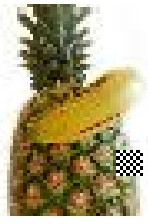

    (a) Training Data                    (b) Training Data                    (c) Test Data

Figure 8: FT trigger on Tiny ImageNet data. Training Data 8a and 8b are from different malicious clients. Test Data 8c is used to test ASR.

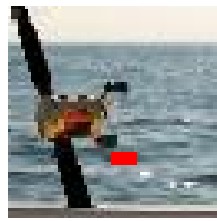 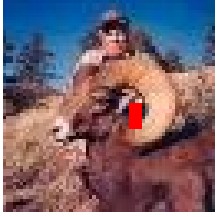 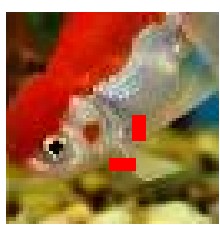

    (a) Training Data                    (b) Training Data                    (c) Test Data

Figure 9: DFT trigger on Tiny ImageNet data. Training Data 9a and 9b are from different malicious clients. Test Data 9c is used to test ASR.

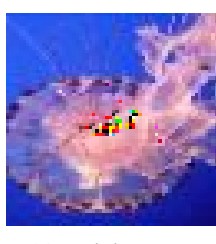 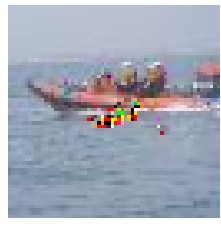 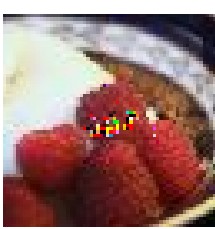

    (a) Training Data                    (b) Training Data                    (c) Test Data

Figure 10: DPOT trigger on Tiny ImageNet data. Training Data 10a and 10b are from different malicious clients. Test Data 10c is used to test ASR.

## F.1    DIFFERENT TYPES OF TRIGGER ON IMAGES

We displayed different types of triggers on images from the Tiny ImageNet dataset in Figures 10, 8, and 9. The pattern of the FT trigger remains consistent across all datasets. The DFT triggers shown in Figure 9 are the same as those used for images from the CIFAR10 dataset, while for the Fashion MNIST and FEMNIST datasets, DFT triggers appear in black.

## F.2    DPOT TRIGGERS ON IMAGES FROM DIFFERENT DATASETS.

We displayed DPOT triggers generated for images from different dataset in Figure 11. Our triggers are in a small size that could not obscure important details of any images.

## F.3    A3FL TRIGGER ON IMAGES OF CIFAR10

In Figure 12, we showed triggers generated by A3FL's methods on images from CIFAR10.

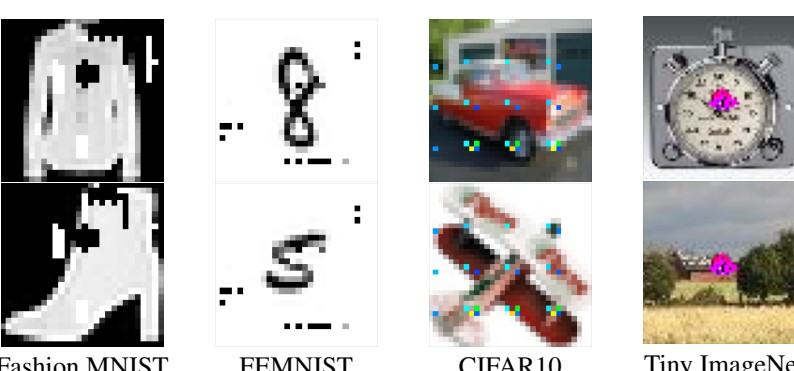

Fashion MNIST  FEMNIST  CIFAR10  Tiny ImageNet

Figure 11: DPOT triggers on images from different datasets.

Figure 12: A3FL trigger on images from CIFAR10.

### F.4 TRIGGER EVOLUTION DURING TRAINING

In Figure 15 and Figure 16, we demonstrated how DPOT trigger changes during the FL training.

In Figure 15, we showed one screenshot of the trigger on a blank background in the same size of the cifar10's figure for every ten global rounds. These trigger screenshots were collected during a DPOT attacking experiment that trains ResNet18 as the global model on the CIFAR-10 dataset, with Trimmed Mean used as the aggregation rule. Figure 13 displays the Main-task Accuracy and Attack Success Rate of the global model over 150 global rounds in this experiment.

Similarly, in Figure 16 we showed one screenshot of the trigger on a blank background in the same size of the Tiny ImageNet's figure for every ten global rounds. These trigger screenshots were collected during a DPOT attacking experiment that trains VGG11 as the global model on the Tiny ImageNet dataset, with Trimmed Mean used as the aggregation rule. Figure 14 displays the Main-task Accuracy and Attack Success Rate of the global model over 100 global rounds in this experiment.

According to Figure 15 and Figure 16, the DPOT trigger does not change drastically over rounds; instead, it develops gradually and coherently. Since the DPOT trigger is optimized based on the global model's parameters, and the global model is in turn influenced by malicious model updates backdoored by the DPOT trigger, the DPOT trigger and the global model form a Markov chain. During training, as the global model evolves coherently and gradually, the states of the DPOT trigger evolve as well in the same pattern.

Global model's accuracy on main/backdoor tasks

Global model's accuracy on main/backdoor tasks

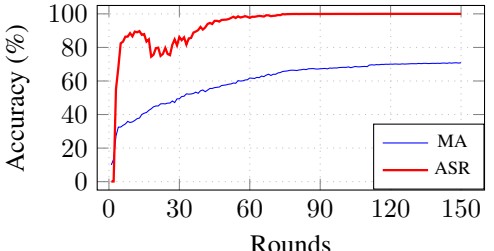

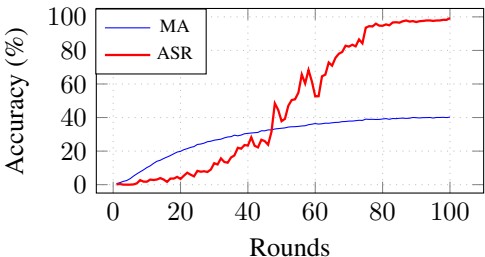

Figure 13: Global model's accuracy in experiment of getting trigger screenshots in Figure 15. (CIFAR10, ResNet18)

Figure 14: Global model's accuracy in experiment of getting trigger screenshots in Figure 16. (Tiny ImageNet, VGG11)

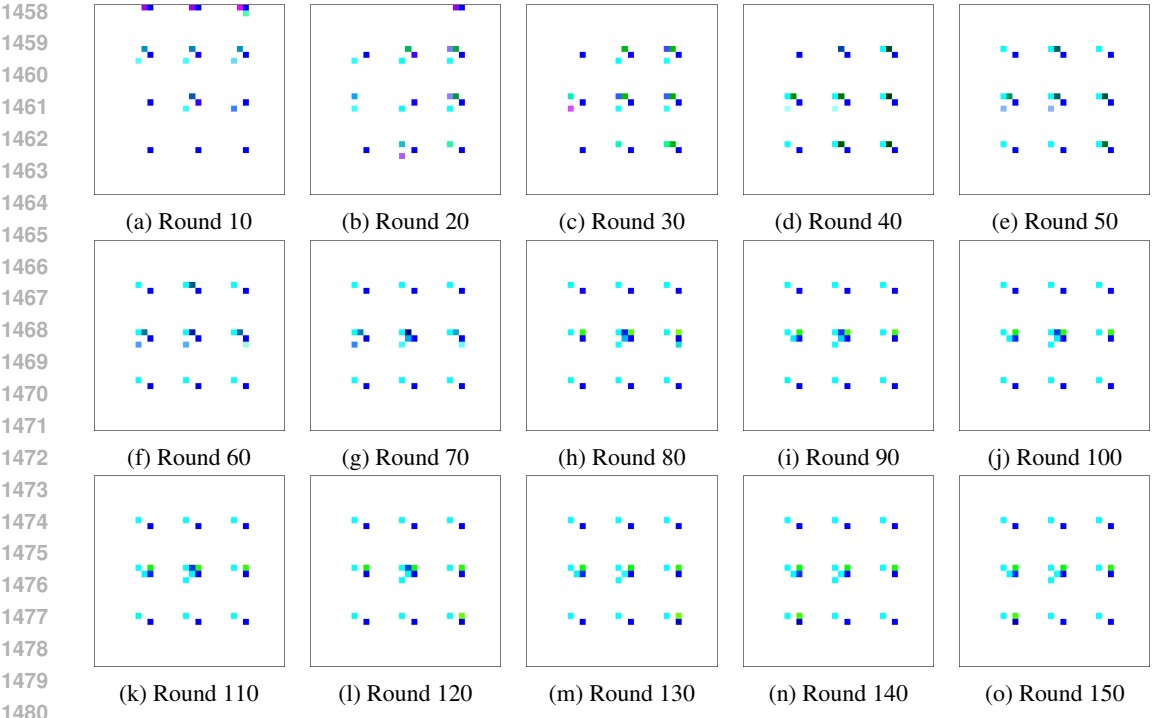

(a) Round 10     (b) Round 20     (c) Round 30     (d) Round 40     (e) Round 50

(f) Round 60     (g) Round 70     (h) Round 80     (i) Round 90     (j) Round 100

(k) Round 110     (l) Round 120     (m) Round 130     (n) Round 140     (o) Round 150

Figure 15: (CIFAR10, ResNet18) DPOT triggers on different rounds.

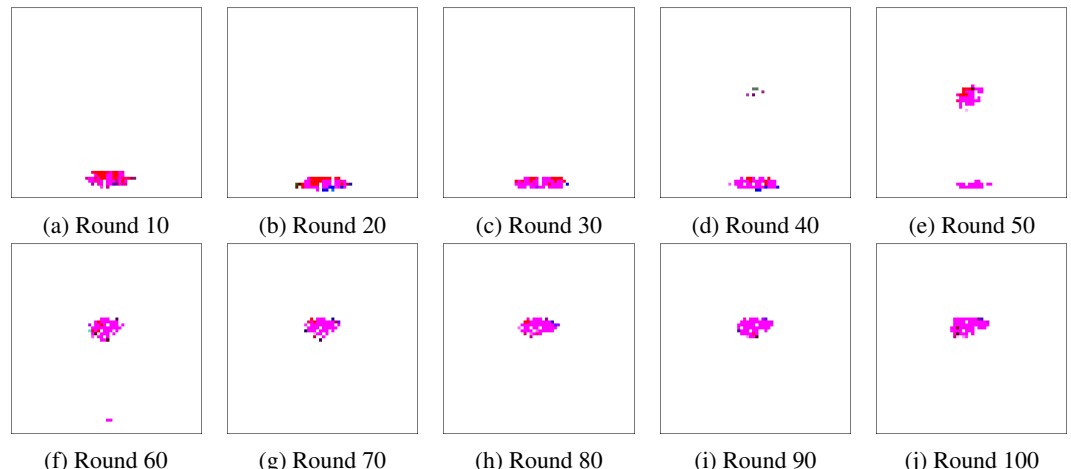

(a) Round 10     (b) Round 20     (c) Round 30     (d) Round 40     (e) Round 50

(f) Round 60     (g) Round 70     (h) Round 80     (i) Round 90     (j) Round 100

Figure 16: (Tiny ImageNet, VGG11) DPOT triggers on different rounds.

