# OpenReview forum: "Concealing Backdoors in Federated Learning by Trigger-Optimized Data Poisoning"
_ICLR.cc/2025/Conference — Submitted to ICLR 2025_

### Official Review · Reviewer_X9XH · 2024-10-27

**Soundness:** 3
**Presentation:** 2
**Contribution:** 2
**Rating:** 3
**Confidence:** 4

**Summary:**

This work proposes a data-poisoning backdoor method with dynamic malicious training objective. Authors argue that current backdoor training algorithms either adopt fixed trigger pattern, or highly relies on model-poisoning assumption. These attacks could fail to escape from current defenses as backdoor updates deviate much from benign ones. Methods based on model poisoning are unrealistic under TEEs. This work thus proposes DPOT to dynamically search for optimal trigger patterns to avoid the detection of defenses mechanisms. Authors further provide empirical results to demonstrate the effectiveness of the method against several defense mechanisms in comparison with other backdoor training algorithms.

**Strengths:**

1. This paper identify a novel scenario, where the existence of TEEs eliminates the possibility of model-poisoning attacks.
2. well-written with clear structure.

**Weaknesses:**

1. This paper claims that the static optimization objective makes current backdoor training algorithms generate updates with distinct difference with benign ones. This could result in the rejection of many backdoor detection methods. However, a recent work [1] finds that adversaries could simply adopt smaller learning rates to train backdoored models. This could craft malicious updates which are statistically close to benign ones, bypassing most defense methods. This raises concern about the necessity about the proposed method. Authors could further conduct experiments under different adversarial settings (different learning rates), and also against the recent work [1].
2. The newly proposed method, DPOT, involves trigger optimization for every global round. And since optimized triggers are different for every global round, will the malicious training for early rounds (with different triggers) helps the training for the last round (the final evaluated trigger)? Otherwise, i cant find reasons for continuously poisoning the FL systems with different triggers.
3. As DPOT introduces extra operations after receiving the global model, time consumed by the optimization procedure could be important. This is because extra processing time could may cause the drop out of malicious clients. I suggest that authors could further discuss on this part.
4. Carefully manipulating the pixel location and pixel value for triggers is not realistic for adversaries to deploy. I dont see a practical setting where adversaries could such precisely control the backdoor pixel location and value. Authors may consider specify this point in the threat model part, otherwise it is inconvincible that the described method is dangerous.
5. The abstract part claims this work provides theoretical justification about the methods. However, i could only find this part in the appendix. I suggest authors could put some of the main results in the main paper.

[1] Li, Songze, and Yanbo Dai. "BackdoorIndicator: Leveraging OOD Data for Proactive Backdoor Detection in Federated Learning." USENIX Security 2024.

**Questions:**

1. could DPOT still outperform other methods under different malicious settings? will it bypass the mentioned work?
2. could the malicious training for early rounds (with different triggers) helps the training for the last round (the final evaluated trigger)?
3. is DPOT practical at least for some settings?

---

> ### Author Response · Authors · 2024-11-24
> **Responses to Weakness 1 (Question 1) and Weakness 2 (Question 2)**
>
> ***Weakness 1**: This paper claims that the static optimization objective makes current backdoor training algorithms generate updates with distinct difference with benign ones. This could result in the rejection of many backdoor detection methods. However, a recent work [1] finds that adversaries could simply adopt smaller learning rates to train backdoored models. This could craft malicious updates which are statistically close to benign ones, bypassing most defense methods. This raises concern about the necessity about the proposed method. Authors could further conduct experiments under different adversarial settings (different learning rates), and also against the recent work [1]. **Question 1**: could DPOT still outperform other methods under different malicious settings? will it bypass the mentioned work?*
>
> **Response**:
> Thank you for bringing up this recent defense work. We conducted experiments with different learning rates to demonstrate DPOT's attack effectiveness against BackdoorIndicator, comparing it to Fixed Pixel-Pattern Triggers (FT).
>
> **Comparison of DPOT and FT's ASR on CIFAR-10 with Non-IID Degree 0.5**
> | Learning Rate | 0.01 | 0.025 | 0.05 |
> | ----| ---- | ---- | ---- |
> | Fixed pixel-pattern (Final ASR) | 10.7 | 23.3 | 26.3 |
> | DPOT (Final ASR) | 100 | 99.9 | 99.9 |
> | DPOT (Avg ASR) | 70.5 | 89.6 |91.2 |
>
> As shown in the table above, the DPOT trigger maintains a significant Final ASR (> 50%) against BackdoorIndicator across different learning rates and outperforms FT. We observe that BackdoorIndicator's defense effectiveness improves with smaller learning rates, consistent with the results in its original paper.
>
> ***Weakness 2**:The newly proposed method, DPOT, involves trigger optimization for every global round. And since optimized triggers are different for every global round, will the malicious training for early rounds (with different triggers) helps the training for the last round (the final evaluated trigger)? Otherwise, i cant find reasons for continuously poisoning the FL systems with different triggers. **Question 2**: could the malicious training for early rounds (with different triggers) helps the training for the last round (the final evaluated trigger)?*
>
> **Response**: Thank you for this insightful suggestion! Optimized triggers differ for each global round, but they share a hidden association due to the Markov chain formed by the global model and optimized triggers. The DPOT trigger is optimized based on the global model’s parameters, and the global model, in turn, is influenced by malicious model updates backdoored by the DPOT trigger. Therefore, early-round triggers impact later-round triggers. To better understand the optimal timing for initiating a DPOT attack, we added experiments to evaluate the DPOT attack's effectiveness at different rounds and frequencies. The results are shown in the following table.
>
> **Final ASR of DPOT when attacking at different round and frequency on CIFAR10**
> | Start round | Interval round  |  FLAIR | FLAME |
> | --- |  --- |--- |--- |
> | 1 | 1 | 62.3| **59.8** |
> | 1 | 5 | 52.9| 53.8 |
> | 1 | 10 | 72.0 | 57.8 |
> | 50 | 1 | 52.1 | 44.9|
> | 50 | 10 | **76.5** | 48.5|
> | 100 | 1 | 54.2 | 50.3|
> | 100 | 10 |69.2 | 54.5|
>
> - Start round: The round that DPOT starts attacking.
> - Interval round: The number of rounds between two adjacent DPOT attacks.
>
> In conclusion, DPOT attack shows better attack effectiveness in lower attacking frequency when against specific defense strategy like FLAIR. FLAIR penalizes clients that are frequently flagged as suspicious by lowering their aggregation weight, reducing their impact on the global model. Malicious clients can regain their normal influence by pausing malicious behavior, allowing the penalty score to gradually decrease. For other defenses that do not use a similar strategy, the effectiveness of the DPOT attack shows insignificant variation across different start rounds and interval rounds.

---

> ### Author Response · Authors · 2024-11-24
> **Responses to Weakness 3, Weakness 4 (Question 3), and Weakness 5**
>
> ***Weakness 3**: As DPOT introduces extra operations after receiving the global model, time consumed by the optimization procedure could be important. This is because extra processing time could may cause the drop out of malicious clients. I suggest that authors could further discuss on this part.*
>
> **Response**: Thanks for this constructive suggestion! Including comparison results in terms of computational overhead has definitely helped to better clarify DPOT's unique advantages. We measured the execution time of DPOT, A3FL, and IBA attacks on the same computational platform, consisting of one NVIDIA A40 GPU core and 200 GB of CPU RAM. The measurement results are presented in the table below.
>
> |     | Overall Elapse Time| Elapsed Time per Epoch | Number of Epochs |  Local Training Elapsed Time |
> | --- |--- |--- |--- |--- |
> | DPOT| 5.05 s | 0.50 s | 10 | 1.23 s |
> | A3FL| 421.04 s| 2.07 s | 200 | 1.23 s |
> | IBA | 16.56 s | 1.59 s | 10 | 1.23 s |
>
> - Overall Elapsed Time: The total execution time for an attacker, from receiving the global model to submitting its training data to the training pipeline in one round.
> - Elapsed Time per Epoch: The time taken for each epoch to optimize a trigger.
> - Number of Epochs: The number of epochs needed to complete trigger optimization. We obtained this data for A3FL and IBA from their default settings for the CIFAR-10 dataset in their open-source projects.
> - Local Training Elapsed Time: The training time for each client. The execution of local training is consistent among all clients, as it is handled within TEEs.
>
> The DPOT attack requires less computational time overall and per epoch, highlighting its efficiency compared to other optimization-based attack methods. The overall elapsed time for a DPOT attack is 4.1 times longer than the local training time. In a distributed FL system with heterogeneous client devices, high-end mobile devices, such as the iPhone 15 Pro, can serve as training platforms. We assume adversaries have access to high-performance computational resources, allowing them to compensate for any computational gap between malicious and benign clients. For instance, the iPhone 15 Pro, with approximately 1 TFLOP GPU performance, is significantly outmatched by an NVIDIA A100 GPU, which delivers up to 312 TFLOPS, demonstrating a 312-fold difference in computational power.
>
>
> ***Weakness 4**:Carefully manipulating the pixel location and pixel value for triggers is not realistic for adversaries to deploy. I dont see a practical setting where adversaries could such precisely control the backdoor pixel location and value. Authors may consider specify this point in the threat model part, otherwise it is inconvincible that the described method is dangerous. **& Question 3**: is DPOT practical at least for some settings?*
>
> **Response**: Thanks for bringing up the discussion about DPOT's deployment. We envision an attack that manipulates (poisons) the training data before it enters the trusted training pipeline, such as at the sensor where the image is initially captured. This method is similar to how to practically deploy other optimized triggers such as A3FL and IBA.
> [1] H. Zhang, J. Jia, J. Chen, L. Lin, and D. Wu. A3fl: Adversarially adaptive backdoor attacks to federated learning. In 36thAdvances in Neural Information Processing Systems, 2024.
> [2] Nguyen, Thuy Dung, et al. "Iba: Towards irreversible backdoor attacks in federated learning." Advances in Neural Information Processing Systems 36 (2024).
>
>
> ***Weakness 5**:The abstract part claims this work provides theoretical justification about the methods. However, i could only find this part in the appendix. I suggest authors could put some of the main results in the main paper.*
>
> **Response**: Thanks for the suggestion! We particularly wanted to include our theoretical justification in the main paper, but were unable to do so due to page limitations and concerns about maintaining a complete narrative of our method.

---

> > ### Comment · Reviewer_X9XH · 2024-11-25
> >
> > Thanks for the additional experimental results. However, I am still not convinced by the reason why early-round triggers facilitate the injection of the final trigger. I hope the authors could work more on this part from either the experimental level or theoretical level to provide clear reasoning. I have decided to keep my initial score.

---

> > > ### Author Response · Authors · 2024-11-28
> > >
> > > Thanks for this critical question. Can we interpret this question as asking whether the DPOT attack is a single-shot or multi-shot attack? A single-shot attack implies that the attack's effectiveness is independent across rounds, meaning each round's performance is unaffected by previous rounds. In contrast, a multi-shot attack suggests that the effectiveness in one round depends on the performance of previous rounds and gradually improves over time.
> > >
> > > To study this property of DPOT, we conduct experiments by initiating the DPOT attack at different rounds and observing whether the ASR at specific subsequent points varies due to the prolonged duration of the attack. A single-shot attack would not exhibit such variations, whereas a multi-shot attack would.
> > >
> > > In the following experiments on the Fashion MNIST dataset, we tested four different attack starting rounds: 1, 200, 250, and 280, with the total number of training rounds being 300. The DPOT attack was conducted in every round after the starting round. Observation rounds were set at 200, 250, 280, and 300 (final).
> > >
> > > **ASR at specific rounds, Trimmed mean**
> > >
> > > | Attack starting round | round 1 | round 200 | round 250 | round 280 | round 300 (Final)|
> > > | ----- | ---- | ---- |---- |---- |---- |
> > > | 1 | 10.0 | 76.27 | 89.52 | 93.6 | 95.64 |
> > > | 200 | - | 49.56 | 84.03 | 91.26 | 93.14 |
> > > | 250 | - | - | 69.47 | 81.04 | 87.86 |
> > > | 280 | - | - | - | 66.75 | 74.41 |
> > >
> > >
> > > **ASR at specific rounds, FLAME**
> > >
> > > | Attack starting round | round 1 | round 200 | round 250 | round 280 | round 300 (Final) |
> > > | ----- | ---- | ---- |---- |---- |---- |
> > > |1|10.0|69.03|93.51| 96.74 | 97.85 |
> > > | 200 |-| 51.41 | 90.55 | 94.6 | 96.04 |
> > > | 250 |-|-| 70.72 | 87.55 | 93.21 |
> > > | 280 |-|-|-| 69.42 | 84.78 |
> > >
> > >
> > > We made the following observations:
> > > - At a specific round, experiments where the DPOT attack started before this point show higher ASR compared to cases with no prior attack. Additionally, an earlier starting round results in a higher ASR at that specific round.
> > > - An earlier attack starting round leads to a higher final-round ASR.
> > > - Optimized triggers exhibit visual similarities with minor variations across different rounds.
> > >
> > > We conclude from the above observations that DPOT exhibits multi-shot attack behavior under these experimental settings. An explanation for this conclusion is that, due to the limited trigger size, purely optimizing triggers is insufficient to achieve the model's optimal misclassification performance on poisoned data (hence, DPOT is not a test-stage attack like FGSM). The optimized trigger aims to find the closest backdoor state for the current global model and gradually guides the global model toward this state by incorporating malicious updates. This process involves continuous attacks with associated triggers, where the association is formed by optimizing the triggers for global models in adjacent states. Thus, starting the attack earlier improves the ASR in subsequent rounds.
> > >
> > > However, DPOT cannot be simply categorized as a multi-shot attack. We also made the following observation in an experiment using the CIFAR-10 dataset with FLAIR applied as a defense.
> > >
> > > **ASR at specific rounds, FLAIR**
> > >
> > > | Attack starting round | round 1 | round 100 | round 140 | round 145 | round 150 (Final)|
> > > | ----- | ---- | ---- |---- |---- |---- |
> > > | 1 | 10.0 | 57.48 | 65.43 | 60.72 | 61.29 |
> > > | 100 | - | 46.49 | 79.54 | 51.0 | 49.24 |
> > > | 140 | - | - | 62.62 | 57.19 | 59.11 |
> > > | 145 | -| - | - | 63.63 | 63.42 |
> > >
> > > The observations are as follows:
> > > - The ASR at different observation rounds is very similar, despite the different attack starting rounds.
> > > - The final-round ASR is not affected by early-round attacks.
> > > - Optimized triggers exhibit visual dissimilarity across different rounds.
> > >
> > > The conclusion based on the above observations is that DPOT behaves as a single-shot attack under this experimental setting. The reason for this is that the model (ResNet18) is large, with a significant number of parameters, resulting in more backdoor states surrounding the model's state. Robust aggregation methods like FLAIR can effectively disrupt the coherence of the global model's convergence path, causing the closest backdoor states of the global model to differ across adjacent rounds. This disruption makes DPOT significantly alter the backdoor optimization objectives accordingly. As a result, the optimized triggers across different rounds become incoherent, and the attack's impact is confined to a single round.
> > >
> > >
> > > Regardless of whether DPOT behaves as a single-shot or multi-shot attack in a given situation, its backdoor optimization objectives always evolve along with the state of the global model, leading to greater attack effectiveness compared to fixing the backdoor objective. We encourage conducting the DPOT attack continuously, not only for the potential of higher ASR but also because the attacker may not know when the final round of training will occur, so they will continually attempt to carry out the attack.

---

### Official Review · Reviewer_5K7y · 2024-11-03

**Soundness:** 3
**Presentation:** 3
**Contribution:** 3
**Rating:** 6
**Confidence:** 5

**Summary:**

This paper addresses a critical issue in federated learning: the threat of backdoor attacks. It proposes a novel attack method called DPOT (Data Poisoning with Optimized Trigger), which leverages trigger optimization to effectively conceal malicious model updates. Specifically, DPOT dynamically adjusts the trigger pattern by identifying the optimal locations and values of the trigger using gradient scores for each pixel. An extensive evaluation on four datasets: Fashion-MNIST, CIFAR-10, FEMNIST, and Tiny ImageNet along with a comprehensive analysis of backdoor defense mechanisms, demonstrates the effectiveness of DPOT.

**Strengths:**

- The evaluation metrics (Final ASR, Average ASR, Main-task Accuracy) are well-defined and provide a thorough assessment of the attack’s effectiveness.
- The analysis of various hyperparameter impacts on attack performance offers valuable insights into DPOT’s behavior.
- A detailed analysis of DPOT’s working principle is provided, distinguishing between the aggregation of backdoored model updates and benign updates.

**Weaknesses:**

- While DPOT effectively hides malicious model updates, the backdoor images still display a visible trigger, which could limit the attack's practicality in real-world applications. Have the authors considered techniques such as IBA [1] or similar methods to make the trigger less detectable? An analysis comparing DPOT's trigger visibility with other state-of-the-art methods, including a discussion on the trade-off between trigger visibility and attack effectiveness, would be valuable.
- DPOT’s use of an adversarial-style trigger is likely a significant factor in its success, which may explain its improved performance over fixed-pixel triggers. However, it would strengthen the paper by providing a detailed comparison with other adversarial trigger methods, such as A3FL, in terms of both attack effectiveness and computational overhead. Conducting experiments that directly compare DPOT with A3FL under the same settings could help clarify DPOT’s unique advantages.
- The paper does not specify the number of pixels in Algorithm 1, which is crucial for the optimization process. How is this pixel count determined in practice?
- The paper lacks a discussion on the potential limitations of DPOT and possible future directions to address them. For instance, the authors could suggest strategies to defend against DPOT, propose enhancements to strengthen existing defense mechanisms, or explore applying the technique to other domains (e.g., text or time-series data) and FL tasks (e.g., regression or reinforcement learning).

**Questions:**

- Could an $L_\infty$ norm be used to constrain the trigger pattern, and how would this impact attack performance?
- How does the optimization process in DPOT compare to other optimization-based attacks like A3FL in terms of training time and convergence?
- How does model performance compare when optimization for trigger-pixel values is omitted (i.e. when selected positions are set to 1)?
- Do all attack scenarios require attackers to participate in only one training round?
- In Figure 6 (page 23), global accuracy on the main task drops between rounds 0 and 40, so the server might not need the contribution from clients and can detect the poisoned model early. How can this be prevented?
- The FEMNIST dataset lacks a dedicated test set. How was a test set generated for this dataset?
- How is the non-iid data distribution considered in attack scenarios?

**References:**

[1] Nguyen, Thuy Dung, et al. "IBA: Towards irreversible backdoor attacks in federated learning." Advances in Neural Information Processing Systems 36 (2024).

---

> ### Author Response · Authors · 2024-11-24
> **Response to Weakness 1 and Weakness 3**
>
> ***Weakness 1**: While DPOT effectively hides malicious model updates, the backdoor images still display a visible trigger, which could limit the attack's practicality in real-world applications. Have the authors considered techniques such as IBA [1] or similar methods to make the trigger less detectable? An analysis comparing DPOT's trigger visibility with other state-of-the-art methods, including a discussion on the trade-off between trigger visibility and attack effectiveness, would be valuable.*
> ***& Weakness 3**: The paper does not specify the number of pixels in Algorithm 1, which is crucial for the optimization process. How is this pixel count determined in practice?*
>
> **Response:**
> Thanks for the suggestion! To quantify the visual stealthiness of a trigger, we use a computer vision model as the judge. In the following experiment, we chose a computer vision model over human vision not only because the model provides an objective assessment, but also because the benign performance of a clean model on poisoned data is significant to adversaries who aim to maintain the functionality of their own models while compromising a competitor's model.
>
> We trained a benign global model on clean data under the same training settings as the victim FL system, using it as the **judge model**. We consider a trigger to have good visual stealthiness if its poisoned data can maintain high benign accuracy on the judge model while showing a high attack success rate (ASR) on the victim global model. The following results explain why the DPOT trigger sizes differ across different datasets.
>
> **Judge model's performance on DPOT poisoned data**
> | | Trigger size| 0 | 25 | 64 | 100 |
> | --- | --- | --- | --- | --- | --- |
> |Fashion-| Benign acc |85.76 |79.32 |76.07 |70.53 |
> |-MNIST|Drop (%)| 0 | -7.5 | **-11.30**| -17.76 |
> |FEMNIST| Benign acc |81.24 |68.11 | 45.12 | 28.39 |
> |--- | Drop (%)|  0 | **-16.16** | -44.46 | -65.05 |
> |CIFAR10| Benign acc |70.81 | 52.98 | 35.90 | 25.06 |
> |--- | Drop (%)| 0 | **-25.18**| -49.30 | -64.61 |
> |Tiny- |Benign acc |43.44 | 42.32 | 35.89 | 29.53 |
> | -ImageNet| Drop (%)|  0 | -2.58 | **-17.38**| -32.02 |
>
> - Benign acc: accuracy of poisoned data being predicted to its original bengin label.
> - Drop (%): Benign acc drop compared to when testing clean data (trigger size is 0) on the Judge model.
>
> Based on the ASR results in Table 3 and Figure 3, we selected the trigger size by balancing the trade-off between attack performance and visual stealthiness—a larger trigger size results in a higher ASR but lower benign accuracy. We set the lower bound for 'Drop' at -30% and the lower bound for 'Final ASR' at 50%, and choose the smallest trigger size that meets both constraints.
>
> **Judge model's and Victim model's performance on poisoned data having different types of optimized triggers (CIFAR10)**
> | Attack |  None | DPOT | IBA | A3FL |
> | ---|---|---|---|---|
> | Benign acc | 70.81 | 52.98 | 7.98 |70.46  |
> | Drop (%) |0 | -25.18 |-88.73  | -0.49 |
> | Final ASR against Trimmed Mean | N/A | 100 | 10.2 | 35.0 |
>
> We compared the visual stealthiness and attack performance of different types of optimized triggers. We reproduced the triggers of IBA and A3FL using their default settings for CIFAR-10, as provided in their open-source projects. For $L_2$-norm constrained triggers like IBA's, the additive perturbations spread across the entire image cause a significant drop in the judge model's benign accuracy, indicating poor visual stealthiness to benign computer vision models. In contrast, for $L_0$-norm constrained triggers, like those used by IBA and DPOT, the judge model’s benign accuracy on poisoned data is only slightly reduced. Compared to A3FL's trigger, the DPOT trigger demonstrates better attack performance, as reflected in its higher Final ASR (Table 6).

---

> ### Author Response · Authors · 2024-11-24
> **Responses to Weakness 2 (Question 2), Weakness 4, Question 1, and Question 3**
>
> ***Weakness 2**: DPOT’s use of an adversarial-style trigger is likely a significant factor in its success, which may explain its improved performance over fixed-pixel triggers. However, it would strengthen the paper by providing a detailed comparison with other adversarial trigger methods, such as A3FL, in terms of both attack effectiveness and computational overhead. Conducting experiments that directly compare DPOT with A3FL under the same settings could help clarify DPOT’s unique advantages. **& Question 2**: How does the optimization process in DPOT compare to other optimization-based attacks like A3FL in terms of training time and convergence?*
>
> **Response**: Thanks for this constructive suggestion! Including comparison results in terms of computational overhead has definitely helped to better clarify DPOT's unique advantages. We measured the execution time of DPOT, A3FL, and IBA attacks on the same computational platform, consisting of one NVIDIA A40 GPU core and 200 GB of CPU RAM. The measurement results are presented in the table below.
>
> |     | Overall Elapse Time| Elapsed Time per Epoch | Number of Epochs |  Local Training Elapsed Time |
> | --- |--- |--- |--- |--- |
> | DPOT| 5.05 s | 0.50 s | 10 | 1.23 s |
> | A3FL| 421.04 s| 2.07 s | 200 | 1.23 s |
> | IBA | 16.56 s | 1.59 s | 10 | 1.23 s |
>
> - Overall Elapsed Time: The total execution time for an attacker, from receiving the global model to submitting its training data to the training pipeline in one round.
> - Elapsed Time per Epoch: The time taken for each epoch to optimize a trigger.
> - Number of Epochs: The number of epochs needed to complete trigger optimization. We obtained this data for A3FL and IBA from their default settings for the CIFAR-10 dataset in their open-source projects.
> - Local Training Elapsed Time: The training time for each client. The execution of local training is consistent among all clients, as it is handled within TEEs.
>
> The DPOT attack requires less computational time overall and per epoch, highlighting its efficiency compared to other optimization-based attack methods. The overall elapsed time for a DPOT attack is 4.1 times longer than the local training time. In a distributed FL system with heterogeneous client devices, high-end mobile devices, such as the iPhone 15 Pro, can serve as training platforms. We assume adversaries have access to high-performance computational resources, allowing them to compensate for any computational gap between malicious and benign clients. For instance, the iPhone 15 Pro, with approximately 1 TFLOP GPU performance, is significantly outmatched by an NVIDIA A100 GPU, which delivers up to 312 TFLOPS, demonstrating a 312-fold difference in computational power.
>
>
> ***Weakness 4**:The paper lacks a discussion on the potential limitations of DPOT and possible future directions to address them. For instance, the authors could suggest strategies to defend against DPOT, propose enhancements to strengthen existing defense mechanisms, or explore applying the technique to other domains (e.g., text or time-series data) and FL tasks (e.g., regression or reinforcement learning).*
>
> **Response**: Thank you for the thoughtful suggestions. A potential limitation of DPOT is that our algorithms are designed for data with a continuous range of values, such as pixel values in images. For data with a discrete range of values, such as text, we have not yet explored potential optimization algorithms. We believe this could be a promising direction for extending the DPOT attack in the future.  We agree that exploring defense strategies against DPOT and proposing enhancements to existing mechanisms could provide valuable insights for the FL security community. We look forward to investigating these areas in future work to further advance the understanding and resilience of federated learning systems.
>
>
> ***Question 1**: Could an $L_\infty$ norm be used to constrain the trigger pattern, and how would this impact attack performance?*
>
> **Response**: The $L_\infty$-norm constrained perturbations appear to be a very interesting idea in creating adversarial example. We look forward to studying its performance in backdoor attacks in future work.
>
> ***Question 3**: How does model performance compare when optimization for trigger-pixel values is omitted (i.e. when selected positions are set to 1)?*
>
> **Response**: During the development of the DPOT attack, we found that combining trigger-pixel value optimization with trigger-pixel placement optimization resulted in better attack effectiveness than using either one alone. Propositions C.2 and C.3 in Appendix C provide theoretical justification for the effectiveness of combining the two optimizations in reducing backdoor loss.

---

> ### Author Response · Authors · 2024-11-24
> **Responses to Question 4, Question 5, and Question 6**
>
> ***Question 4**: Do all attack scenarios require attackers to participate in only one training round?*
>
> **Response**: In our work, each client participate in FL training for each round, so malicious client can conduct attacking in each round. We added experiments to evaluate DPOT attack's extensive capability in attacking at different round and different frequency. Results are shown in the following table.
>
> **Final ASR of DPOT when attacking at different round and frequency on CIFAR10**
> | Start round | Interval round  |  FLAIR | FLAME |
> | --- |  --- |--- |--- |
> | 1 | 1 | 62.3| **59.8** |
> | 1 | 5 | 52.9| 53.8 |
> | 1 | 10 | 72.0 | 57.8 |
> | 50 | 1 | 52.1 | 44.9|
> | 50 | 10 | **76.5** | 48.5|
> | 100 | 1 | 54.2 | 50.3|
> | 100 | 10 |69.2 | 54.5|
>
> - Start round: The round that DPOT starts attacking.
> - Interval round: The number of rounds beween two adjacent DPOT attacks.
>
> In conclusion, DPOT attack shows better attack effectiveness in lower attacking frequency when against specific defense strategy like FLAIR. FLAIR penalizes clients that are frequently flagged as suspicious by lowering their aggregation weight, reducing their impact on the global model. Malicious clients can regain their normal influence by pausing malicious behavior, allowing the penalty score to gradually decrease. For other defenses that do not use similar strategy, the effectiveness of the DPOT attack shows insignificant variation across different start rounds and interval rounds.
>
>
> ***Question 5**: In Figure 6 (page 23), global accuracy on the main task drops between rounds 0 and 40, so the server might not need the contribution from clients and can detect the poisoned model early. How can this be prevented?*
>
> **Response**: Thank you for your careful observation. We may not be the best authority to comment on Flip's performance, as we are not its developers. However, based on our experiments shown in Figure 6, we observed that the MA of the global model decreases at the early stage, regardless of whether the system is under attack (represented by the "Continuous" line) or not (represented by the "Single shot" line). Therefore, we were uncertain whether these drops could serve as a potential signal of an attack. Additionally, since Flip's implementation uses a pretrained model for initialization, could we interpret any subsequent state of the global model during training as a new "pretrained" model that begins the training process from scratch? If so, the benign global model at round 40 might not differ in state or meaning from that at round 0, and our attack conclusions should be applicable to both.
>
>
> ***Question 6**: The FEMNIST dataset lacks a dedicated test set. How was a test set generated for this dataset?*
>
> **Response**: We downloaded FEMNIST dataset from the open-source github repository of the project "LEAF: A Benchmark for Federated Settings". We followed the instructions in their FEMNIST subdirectory and used their suggested default argument value “-t sample” to partition each user's samples into train-test groups.

---

> ### Author Response · Authors · 2024-11-24
> **Response to Question 7**
>
> ***Question 7**: How is the non-iid data distribution considered in attack scenarios?*
>
> **Response**: Thank you for this suggestion. All results in our paper are trained under non-IID settings, with the non-IID degree set to 0.5. We implemented experiments to study the impact of different non-IID degrees on the performance of DPOT, FT, and DFT attacks. We evaluated Final ASR and Avg ASR to compare attack effectiveness, and Main-task Accuracy to assess the stealthiness of all backdoor attacks. Other attack and FL training settings are consistent with those in the main body of the paper. The results are shown in the following tables.
>
> ***Final ASR***
> | Non-iid degree |  0 |--- | ---|  0.2 | --- | ---| 0.5 | --- |--- | 0.8 |--- | ---|
> |----|----|----|----|----|----|----|----|----|----|----|----|----|
> |   | Ours | FT | DFT | Ours | FT | DFT | Ours | FT | DFT | Ours | FT | DFT |
> | Fedavg |**100.0** |**100.0**| 99.7|**100.0** |**100.0**| 99.8| **100.0** |**100.0** |92.5 |**100.0** |**100.0**| 99.3|
> | Median |**100.0** |99.4| 72.2|**100.0** |99.0| 66.8|**100.0** |81.3 |72.0 |**99.3** |99.2| 33.0|
> | Trimmed Mean |**100.0** |99.4| 73.6|**100.0** |99.8| 86.8|**100.0** |94.8 |38.3 |**100.0** |96.0| 65.8|
> | RobustLR | **100.0**|**100.0**| 99.8|**100.0**|**100.0**| 99.8| **100.0**|**100.0** |**100.0** | **100.0**|**100.0**| 99.2|
> | RFA |**100.0** |**100.0**| 97.0|**100.0** |99.8| 95.5|**100.0**|**100.0** |98.2 |**100.0** |**100.0**| 98.5|
> | FLAIR |**49.7** |21.2| 12.9|**65.4** |29.3| 15.8| **62.3** |15.1 |9.7 |**19.9** |12.6| 5.1|
> | FLCert |**99.9** |99.7| 77.1|**100.0** |96.7| 54.1|**99.2** |92.6 |33.7 |**100.0** |92.1| 19.4|
> | FLAME |**46.5** |15.7| 15.1|**57.6** |16.7| 16.2|  **59.8**| 19.0 | 16.8 |**100.0** |8.5| 26.5|
> | FoolsGold |**100.0** |**100.0**| 99.9|**100.0** |**100.0**| 99.6|**100.0** |**100.0** |94.1 |**100.0** |**100.0**| 99.9|
> | Multi-Krum |**100.0** |**100.0**| 95.2|**100.0** |**100.0**| 18.9|**100.0**|**100.0** |99.9|**100.0** |**100.0**| **100.0**|
>
> ***Average ASR***
> | Non-iid degree |  0 |--- | ---|  0.2 | --- | ---| 0.5 | --- |--- | 0.8 |--- | ---|
> |----|----|----|----|----|----|----|----|----|----|----|----|----|
> |   | Ours | FT | DFT | Ours | FT | DFT | Ours | FT | DFT | Ours | FT | DFT |
> | Fedavg |**98.1** |94.6| 81.5|**97.9** |90.8| 80.9|**98.5** |88.1 |50.2 |**98.3** |94.2| 75.5|
> | Median |**95.9** |73.0| 39.2|**95.3** |79.5| 38.1|**96.1** |46.6 |41.8 |**89.0** |59.6| 19.6|
> | Trimmed Mean |**95.3** |75.6| 42.6|**96.7** |80.5| 53.7|**88.6** |59.4 |22.6 |**93.3** |49.4| 33.6|
> | RobustLR |**98.5** |95.0| 79.5| **98.6**|92.9| 81.7| **98.6** |94.4 |87.4 |**98.1** |95.1| 71.0|
> | RFA |**95.8** |86.6| 61.4|**97.5** |87.9| 61.5|**97.8** |81.0 |55.3 |**98.0** |85.8| 58.6|
> | FLAIR |**56.7** |27.7| 12.1|**64.9** |28.2| 14.5| **50.7** |13.6 |10.1 |**21.0** |9.3| 5.4|
> | FLCert |**96.7** |71.0| 40.4|**95.0** |74.7| 31.7|**88.3** |59.5 |21.3 |**96.2** |52.2| 13.7|
> | FLAME |**63.0** |38.8| 26.0|**56.1** |28.3| 21.4| **56.0** | 18.2 | 14.4 | **73.8**|21.1| 37.8|
> | FoolsGold |**98.4** |93.9| 81.4|**97.8** |92.6| 81.0|**98.5** |88.4 |53.0 |**98.7**|98.0| 80.0|
> | Multi-Krum |91.4 |**98.2**| 58.4|**98.6** |86.6| 15.1|**98.7**|**98.7** |82.8 |96.2 |**98.7**| 70.4|
>
> ***Main-task Accuracy***
> | Non-iid degree |  0 |--- | ---|  0.2 | --- | ---| 0.5 | --- |--- | 0.8 |--- | ---|
> |----|----|----|----|----|----|----|----|----|----|----|----|----|
> |   | Ours | FT | DFT | Ours | FT | DFT | Ours | FT | DFT | Ours | FT | DFT |
> | Fedavg |74.5 |74.5| 74.9|74.4 |74.3| 75.0|70.7 |70.4 |71.4 |55.9 |55.5| 56.2|
> | Median |74.5 |75.5| 74.3|73.8 |74.3| 74.9|69.1 |69.8 |69.7 |53.4 |52.9| 54.5|
> | Trimmed Mean |75.0 |74.3| 74.5|74.4 |75.2| 74.3|70.4 |70.2 |70.8 |52.7 |53.5| 54.5|
> | RobustLR |75.3 |75.1| 75.1|75.3 |74.4| 74.4| 70.4 |70.4 |70.4 |55.2 |55.8| 55.4|
> | RFA |74.3 |74.8| 75.1|75.0 |75.8| 74.4|70.7 |70.3 |70.8 |56.4|56.2| 55.2|
> | FLAIR |73.7 |73.4| 73.6|73.9 |72.7| 73.1| 70.6 |71.0 |70.4 |55.3 |52.5| 52.1|
> | FLCert |74.6 |74.0| 74.7|74.1 |74.8| 74.0|70.0 |69.8 |70.4 |53.9 |53.5| 53.3|
> | FLAME |73.2 |72.1| 73.0|72.9 |73.5| 73.5| 70.3 | 70.9 | 70.9 |56.1 |56.2| 57.5|
> | FoolsGold |74.1 |74.5| 74.5|74.6 |74.3| 74.5|71.0 |71.2 |71.7 |57.0 |55.1| 55.1|
> | Multi-Krum |73.8 |73.6| 73.1|73.4 |73.2| 73.0|62.9|63.2| 60.8|38.7 |41.0| 38.4|
>
> It can be observed from the last table that different non-IID degrees result in different main-task accuracies. A smaller non-IID degree indicates that the data distribution is closer to an IID distribution, with a non-IID degree of 0 representing an IID distribution. The DPOT attack consistently exhibits better attack effectiveness than FT and DFT, performing well across different non-IID degree settings.

---

> > ### Comment · Reviewer_5K7y · 2024-11-25
> >
> > I appreciate your response and the additional experiments you conducted. They have resolved most of my concerns, so I’ve adjusted my score upward accordingly.

---

> > > ### Author Response · Authors · 2024-11-28
> > >
> > > Thank you for your positive feedback and for increasing the score of our submission. We appreciate the time you took to review our work and are grateful for your constructive comments, which have helped improve the paper.

---

### Official Review · Reviewer_hY2e · 2024-11-03

**Soundness:** 2
**Presentation:** 3
**Contribution:** 2
**Rating:** 5
**Confidence:** 4

**Summary:**

In this work, the authors propose DPOT, a novel backdoor attack method relying solely on data poisoning in federated learning (FL). DPOT dynamically adjusts the backdoor objective to conceal malicious clients' model updates among benign ones, allowing the global model to aggregate them even when protected by state-of-the-art defenses.

**Strengths:**

1.	This paper tackles a challenging FL backdoor attack scenario, where the FL system employs a trusted execution environment to restrict clients from freely modifying the objective function, limiting attackers to data poisoning as their only option for inserting backdoors.
2.	The authors validate DPOT’s effectiveness through extensive experiments and offer theoretical insights to explain its success.
3.	The paper is well-written, clearly conveying both the workings of the proposed method and the reasoning behind it.In this work, the authors propose DPOT, a novel backdoor attack method relying solely on data poisoning in federated learning (FL). DPOT dynamically adjusts the backdoor objective to conceal malicious clients' model updates among benign ones, allowing the global model to aggregate them even when protected by state-of-the-art defenses.

**Weaknesses:**

1.	DPOT’s core concept of using a UAP as a trigger (even though the UAP changes dynamically each round) lacks novelty; similar approaches have already been explored in methods like A3FL and IBA.
2.	Could the authors clarify why A3FL shows a performance far below the values reported in the original paper, even underperforming compared to FT and DFT? This result appears counterintuitive. Also, A3FL should be given more emphasis in the main experiment, rather than being placed in the Appendix.
3.	In Table 5, why is the trigger size for Fashion MNIST set larger than for CIFAR-10 and even equal to Tiny ImageNet?
4.	A comparison between DPOT and IBA (IBA: Towards Irreversible Backdoor Attacks in Federated Learning) would be beneficial, as IBA optimizes the trigger generator, which intuitively offers greater room for enhancement and may outperform all baselines considered here.
5.	In terms of defenses, it would be useful to include more recent approaches, such as BackdoorIndicator (Usenix Security ‘24).
6.	Would optimizing only the Trigger-Pixel Values without limiting Trigger-Pixel Placements lead to better attack performance?
7.	Can DPOT withstand adaptive defenses, where defenders anticipate the use of DPOT and design customized countermeasures?
8.	The impact of different levels of Non-IID on DPOT’s attack performance is not evaluated.

**Questions:**

My questions are included in Weaknesses.

---

> ### Author Response · Authors · 2024-11-24
> **Responses to Weakness 1 and Weakness 2**
>
> ***Weakness 1**: DPOT’s core concept of using a UAP as a trigger (even though the UAP changes dynamically each round) lacks novelty; similar approaches have already been explored in methods like A3FL and IBA.*
>
> **Response**: In addition to its outstanding attack effectiveness compared to existing attacks such as A3FL and IBA, DPOT makes two important contributions to the FL security community that differentiate it from other attacks:
> - DPOT explicitly indicates and proves that FL is highly vulnerable to pure data-poisoning attacks. The absence of this indication in existing works can cause the community to underestimate the harm of pure data-poisoning attacks and hinder the development of more advanced security strategies outside of TEEs.
> - DPOT is the first to proposes a method to generate $L_0$-norm constrained trigger with free shape and placement while being able to specify its exact size, which greatly increases the $L_0$-norm constrained triggers' attack effectiveness. The development of optimizing $L_0$-norm constrained triggers should be encouraged in the same way as $L_2$-norm constrained perturbations.
>
>
> ***Weakness 2**: Could the authors clarify why A3FL shows a performance far below the values reported in the original paper, even underperforming compared to FT and DFT? This result appears counterintuitive. Also, A3FL should be given more emphasis in the main experiment, rather than being placed in the Appendix.*
>
> **Response**: Thanks for this suggestion. A3FL is a very important pioneering work that used optimized $L_0$-norm constrained triggers to attack FL. We appreciates the important step that A3FL has taken for the FL backdoor attacks and respectfully included it in our main comparison baselines. The results of the A3FL experiments were initially included in the main paper, but we adjusted the text at the last minute due to page limitations and paragraph length considerations. We will readjust the text to include it in the main paper. In our experiments, A3FL shows different performance from its original paper because of two reasons:
> - As A3FL claims in their contribution,  "we predict the movement of the global model by assuming that the server can access the backdoor trigger and train the global model to directly unlearn the trigger. We adaptively optimize the backdoor trigger to make it survive this adversarial global model." While in DPOT's threat model and experiment settings, we conservatively assume that the server doesn't possess knowledge about the trigger and so is unable to implement unlearning process on the global model. We believe that both conservative and advancing defense scenarios are equally important in research value.
> - Our FL training configurations (Appendix D.2) and attack's Default settings (Table 5) are different from A3FL's paper (their section 4.1), which may lead to different performance when evaluating the same attacking implementation.

---

> ### Author Response · Authors · 2024-11-24
> **Response to Weakness 3**
>
> ***Weakness 3**: In Table 5, why is the trigger size for Fashion MNIST set larger than for CIFAR-10 and even equal to Tiny ImageNet?*
>
> **Response:**
> Thanks for this suggestion! To quantify the visual stealthiness of a trigger, we use a computer vision model as the judge. In the following experiment, we chose a computer vision model over human vision not only because the model provides an objective assessment, but also because the benign performance of a clean model on poisoned data is significant to adversaries who aim to maintain the functionality of their own models while compromising a competitor's model.
>
> We trained a benign global model on clean data under the same training settings as the victim FL system, using it as the **judge model**. We consider a trigger to have good visual stealthiness if its poisoned data can maintain high benign accuracy on the judge model while showing a high attack success rate (ASR) on the victim global model. The following results explain why the DPOT trigger sizes differ across different datasets.
>
> **Judge model's performance on DPOT poisoned data**
> | | Trigger size| 0 | 25 | 64 | 100 |
> | --- | --- | --- | --- | --- | --- |
> |Fashion-| Benign acc |85.76 |79.32 |76.07 |70.53 |
> |-MNIST|Drop (%)| 0 | -7.5 | **-11.30**| -17.76 |
> |FEMNIST| Benign acc |81.24 |68.11 | 45.12 | 28.39 |
> |--- | Drop (%)|  0 | **-16.16** | -44.46 | -65.05 |
> |CIFAR10| Benign acc |70.81 | 52.98 | 35.90 | 25.06 |
> |--- | Drop (%)| 0 | **-25.18**| -49.30 | -64.61 |
> |Tiny- |Benign acc |43.44 | 42.32 | 35.89 | 29.53 |
> | -ImageNet| Drop (%)|  0 | -2.58 | **-17.38**| -32.02 |
>
> - Benign acc: accuracy of poisoned data being predicted to its original bengin label.
> - Drop (%): Benign acc drop compared to when testing clean data (trigger size is 0) on the Judge model.
>
> Based on the ASR results in Table 3 and Figure 3, we selected the trigger size by balancing the trade-off between attack performance and visual stealthiness—a larger trigger size results in a higher ASR but lower benign accuracy. We set the lower bound for 'Drop' at -30% and the lower bound for 'Final ASR' at 50%, and choose the smallest trigger size that meets both constraints.
>
> **Judge model's and Victim model's performance on poisoned data having different types of optimized triggers (CIFAR10)**
> | Attack |  None | DPOT | IBA | A3FL |
> | ---|---|---|---|---|
> | Benign acc | 70.81 | 52.98 | 7.98 |70.46  |
> | Drop (%) |0 | -25.18 |-88.73  | -0.49 |
> | Final ASR against Trimmed Mean | N/A | 100 | 10.2 | 35.0 |
>
> We compared the visual stealthiness and attack performance of different types of optimized triggers. We reproduced the triggers of IBA and A3FL using their default settings for CIFAR-10, as provided in their open-source projects. For $L_2$-norm constrained triggers like IBA's, the additive perturbations spread across the entire image cause a significant drop in the judge model's benign accuracy, indicating poor visual stealthiness to benign computer vision models. In contrast, for $L_0$-norm constrained triggers, like those used by IBA and DPOT, the judge model’s benign accuracy on poisoned data is only slightly reduced. Compared to A3FL's trigger, the DPOT trigger demonstrates better attack performance, as reflected in its higher Final ASR (Table 6).

---

> ### Author Response · Authors · 2024-11-24
> **Responses to Weakness 4, Weakness 5, Weakness 6, and Weakness 7**
>
> **Weakness 4**:A comparison between DPOT and IBA (IBA: Towards Irreversible Backdoor Attacks in Federated Learning) would be beneficial, as IBA optimizes the trigger generator, which intuitively offers greater room for enhancement and may outperform all baselines considered here.
>
> **Response**:
> Thank you for mentioning related work IBA, which is a representative study using an $L_2$-norm constrained trigger to attack an FL system. We compared the attack results of IBA with our method, as shown in the table below.
>
> **Comaprison results with IBA attack on CIFAR10**
> | Measure | Final ASR| -------- | Average ASR| -------- | Main-task Acc |-------- |
> | --- | --- | --- | --- | --- | --- | --- |
> |  | IBA | Ours | IBA | Ours | IBA | Ours |
> | FedAvg | 18.5| **100** | 26.0 |**98.5** | 69.3 |70.7 |
> | Median | 21.8 | **100** | 14.2  |**96.1** | 69.8 |69.1 |
> | Trimmed Mean | 10.2 | **100** | 12.2 |**88.6** | 69.7 |70.4 |
> | RobustLR | 32.8 |**100**  | 33.7 |**98.6** | 70.3 |70.1 |
> | RFA | 9.0 | **100** | 9.5 |**97.8** | 70.4 | 70.7|
> | FLAIR | 0.1 | **62.3** | 3.7 |**50.7** | 70.5 |70.6 |
> | FLAME | 3.9 |**59.8**  | 25.1 | **56.0**| 68.7 |70.3 |
> | FoolsGold | 14.8 |**100**  | 13.4 |**98.5**| 70.1  |71.0 |
> | FLCert | 4.0 | **99.2** | 4.2 |**88.3** | 69.3 |70.0 |
> | Multi-Krum | 0.3 |**100**  |4.4  |**98.7** |64.3  |63.0 |
>
> In summary, our method demonstrates better attack effectiveness than IBA's method. To explain how we reproduced IBA's results, we integrated the entire attacker implementation from IBA's open-source project into our FL training framework and disabled any model-poisoning techniques to match the TEE settings in our threat model. We faithfully used their attacker configuration for the CIFAR-10 dataset. All experiments were conducted under the same default settings listed in Table 5, with FL training configurations aligned with Appendix D.2.
>
>
>
> **Weakness 5**:In terms of defenses, it would be useful to include more recent approaches, such as BackdoorIndicator (Usenix Security ‘24).
>
> **Response**: Thank you for bring up this recent defense work. We conducted experiments with different learning rates to demonstrate DPOT's attack effectiveness against BackdoorIndicator, comparing it to Fixed Pixel-Pattern Triggers (FT).
>
> **Comparison of DPOT and FT's ASR on CIFAR-10 with Non-IID Degree 0.5**
> | Learning Rate | 0.01 | 0.025 | 0.05 |
> | ----| ---- | ---- | ---- |
> | Fixed pixel-pattern (Final ASR) | 10.7 | 23.3 | 26.3 |
> | DPOT (Final ASR) | 100 | 99.9 | 99.9 |
> | DPOT (Avg ASR) | 70.5 | 89.6 |91.2 |
>
> As shown in the table above, the DPOT trigger maintains a significant Final ASR (> 50%) against BackdoorIndicator across different learning rates and outperforms FT. We observe that BackdoorIndicator's defense effectiveness improves with smaller learning rates, consistent with the results in its original paper.
>
>
> **Weakness 6**:Would optimizing only the Trigger-Pixel Values without limiting Trigger-Pixel Placements lead to better attack performance?
>
> **Response**: During the development of the DPOT attack, we found that combining trigger-pixel value optimization with trigger-pixel placement optimization resulted in better attack effectiveness than using either one alone. Propositions C.2 and C.3 in Appendix C provide theoretical justification for the effectiveness of combining the two optimizations in reducing backdoor loss.
>
> **Weakness 7**: Can DPOT withstand adaptive defenses, where defenders anticipate the use of DPOT and design customized countermeasures?
>
> **Response**: Thank you for this insightful suggestion about adaptive defense. In the advancing defense scenario, defenders might have information about the DPOT trigger in some rounds and attempt to unlearn it from the global model. We look forward to exploring this idea in future work.

---

> ### Author Response · Authors · 2024-11-24
> **Response to Weakness 8**
>
> **Weakness 8**: The impact of different levels of Non-IID on DPOT’s attack performance is not evaluated.
>
> **Response**: Thank you for this suggestion. All results in our paper are trained under non-IID settings, with the non-IID degree set to 0.5. We implemented experiments to study the impact of different non-IID degrees on the performance of DPOT, FT, and DFT attacks. We evaluated Final ASR and Avg ASR to compare attack effectiveness, and Main-task Accuracy to assess the stealthiness of all backdoor attacks. Other attack and FL training settings are consistent with those in the main body of the paper. The results are shown in the following tables.
>
> ***Final ASR***
> | Non-iid degree |  0 |--- | ---|  0.2 | --- | ---| 0.5 | --- |--- | 0.8 |--- | ---|
> |----|----|----|----|----|----|----|----|----|----|----|----|----|
> |   | Ours | FT | DFT | Ours | FT | DFT | Ours | FT | DFT | Ours | FT | DFT |
> | Fedavg |**100.0** |**100.0**| 99.7|**100.0** |**100.0**| 99.8| **100.0** |**100.0** |92.5 |**100.0** |**100.0**| 99.3|
> | Median |**100.0** |99.4| 72.2|**100.0** |99.0| 66.8|**100.0** |81.3 |72.0 |**99.3** |99.2| 33.0|
> | Trimmed Mean |**100.0** |99.4| 73.6|**100.0** |99.8| 86.8|**100.0** |94.8 |38.3 |**100.0** |96.0| 65.8|
> | RobustLR | **100.0**|**100.0**| 99.8|**100.0**|**100.0**| 99.8| **100.0**|**100.0** |**100.0** | **100.0**|**100.0**| 99.2|
> | RFA |**100.0** |**100.0**| 97.0|**100.0** |99.8| 95.5|**100.0**|**100.0** |98.2 |**100.0** |**100.0**| 98.5|
> | FLAIR |**49.7** |21.2| 12.9|**65.4** |29.3| 15.8| **62.3** |15.1 |9.7 |**19.9** |12.6| 5.1|
> | FLCert |**99.9** |99.7| 77.1|**100.0** |96.7| 54.1|**99.2** |92.6 |33.7 |**100.0** |92.1| 19.4|
> | FLAME |**46.5** |15.7| 15.1|**57.6** |16.7| 16.2|  **59.8**| 19.0 | 16.8 |**100.0** |8.5| 26.5|
> | FoolsGold |**100.0** |**100.0**| 99.9|**100.0** |**100.0**| 99.6|**100.0** |**100.0** |94.1 |**100.0** |**100.0**| 99.9|
> | Multi-Krum |**100.0** |**100.0**| 95.2|**100.0** |**100.0**| 18.9|**100.0**|**100.0** |99.9|**100.0** |**100.0**| **100.0**|
>
> ***Average ASR***
> | Non-iid degree |  0 |--- | ---|  0.2 | --- | ---| 0.5 | --- |--- | 0.8 |--- | ---|
> |----|----|----|----|----|----|----|----|----|----|----|----|----|
> |   | Ours | FT | DFT | Ours | FT | DFT | Ours | FT | DFT | Ours | FT | DFT |
> | Fedavg |**98.1** |94.6| 81.5|**97.9** |90.8| 80.9|**98.5** |88.1 |50.2 |**98.3** |94.2| 75.5|
> | Median |**95.9** |73.0| 39.2|**95.3** |79.5| 38.1|**96.1** |46.6 |41.8 |**89.0** |59.6| 19.6|
> | Trimmed Mean |**95.3** |75.6| 42.6|**96.7** |80.5| 53.7|**88.6** |59.4 |22.6 |**93.3** |49.4| 33.6|
> | RobustLR |**98.5** |95.0| 79.5| **98.6**|92.9| 81.7| **98.6** |94.4 |87.4 |**98.1** |95.1| 71.0|
> | RFA |**95.8** |86.6| 61.4|**97.5** |87.9| 61.5|**97.8** |81.0 |55.3 |**98.0** |85.8| 58.6|
> | FLAIR |**56.7** |27.7| 12.1|**64.9** |28.2| 14.5| **50.7** |13.6 |10.1 |**21.0** |9.3| 5.4|
> | FLCert |**96.7** |71.0| 40.4|**95.0** |74.7| 31.7|**88.3** |59.5 |21.3 |**96.2** |52.2| 13.7|
> | FLAME |**63.0** |38.8| 26.0|**56.1** |28.3| 21.4| **56.0** | 18.2 | 14.4 | **73.8**|21.1| 37.8|
> | FoolsGold |**98.4** |93.9| 81.4|**97.8** |92.6| 81.0|**98.5** |88.4 |53.0 |**98.7**|98.0| 80.0|
> | Multi-Krum |91.4 |**98.2**| 58.4|**98.6** |86.6| 15.1|**98.7**|**98.7** |82.8 |96.2 |**98.7**| 70.4|
>
> ***Main-task Accuracy***
> | Non-iid degree |  0 |--- | ---|  0.2 | --- | ---| 0.5 | --- |--- | 0.8 |--- | ---|
> |----|----|----|----|----|----|----|----|----|----|----|----|----|
> |   | Ours | FT | DFT | Ours | FT | DFT | Ours | FT | DFT | Ours | FT | DFT |
> | Fedavg |74.5 |74.5| 74.9|74.4 |74.3| 75.0|70.7 |70.4 |71.4 |55.9 |55.5| 56.2|
> | Median |74.5 |75.5| 74.3|73.8 |74.3| 74.9|69.1 |69.8 |69.7 |53.4 |52.9| 54.5|
> | Trimmed Mean |75.0 |74.3| 74.5|74.4 |75.2| 74.3|70.4 |70.2 |70.8 |52.7 |53.5| 54.5|
> | RobustLR |75.3 |75.1| 75.1|75.3 |74.4| 74.4| 70.4 |70.4 |70.4 |55.2 |55.8| 55.4|
> | RFA |74.3 |74.8| 75.1|75.0 |75.8| 74.4|70.7 |70.3 |70.8 |56.4|56.2| 55.2|
> | FLAIR |73.7 |73.4| 73.6|73.9 |72.7| 73.1| 70.6 |71.0 |70.4 |55.3 |52.5| 52.1|
> | FLCert |74.6 |74.0| 74.7|74.1 |74.8| 74.0|70.0 |69.8 |70.4 |53.9 |53.5| 53.3|
> | FLAME |73.2 |72.1| 73.0|72.9 |73.5| 73.5| 70.3 | 70.9 | 70.9 |56.1 |56.2| 57.5|
> | FoolsGold |74.1 |74.5| 74.5|74.6 |74.3| 74.5|71.0 |71.2 |71.7 |57.0 |55.1| 55.1|
> | Multi-Krum |73.8 |73.6| 73.1|73.4 |73.2| 73.0|62.9|63.2| 60.8|38.7 |41.0| 38.4|
>
> It can be observed from the last table that different non-IID degrees result in different main-task accuracies. A smaller non-IID degree indicates that the data distribution is closer to an IID distribution, with a non-IID degree of 0 representing an IID distribution. The DPOT attack consistently exhibits better attack effectiveness than FT and DFT, performing well across different non-IID degree settings.

---

> > ### Comment · Reviewer_hY2e · 2024-11-25
> >
> > Thank you for the responses to my concerns. I have decided to maintain my initial score.

---

> > > ### Author Response · Authors · 2024-11-28
> > >
> > > Thank you for not reducing the score and for your thoughtful feedback. If you have any additional questions or concerns, we would greatly appreciate the opportunity to address them and further clarify any points.

---

### Official Review · Reviewer_qg3k · 2024-11-04

**Soundness:** 2
**Presentation:** 3
**Contribution:** 2
**Rating:** 5
**Confidence:** 5

**Summary:**

This paper introduces DPOT, a backdoor attack strategy that optimizes the trigger pattern to combine with data instead of using a fixed pattern using multi-object optimization. Unlike previous attacks that relied on fixed patterns or model manipulation, DPOT dynamically adjusts trigger placements and values to blend malicious updates with benign ones, effectively bypassing standard defenses. Experimental results show DPOT’s superior attack success rate across various datasets, demonstrating both high effectiveness and stealthiness in maintaining main-task accuracy​.

**Strengths:**

- This paper is easy to follow with a good structure. The method is clear and sounds.
- This paper has comprehensive evaluations across diverse datasets and defenses.
- The using of dynamic trigger optimization is interesting.

**Weaknesses:**

1. The trigger seems to be very noticeable, especially when the trigger size is high, which can not be effective against post-defense such as Neural Cleanse. The suggestion for the trigger size is vague and does not contain specific number ranges (Appendix D.3), which is hard to apply in real settings. Could the author argue more about it?
2. The authors did not well discuss the difference between this method with related works using optimized triggers nor comparing them [1][2]. The idea of optimizing triggers is somewhat similar to adversarial attack methods like the Basic Iterative Method (BIM) and Fast Gradient Sign Method (FGSM) in terms of iterative gradient-based optimization, but the discussion is missing.
3. The results in Table 8 and Table 9 raise suspicion that in existing attack papers, the MA of CIFAR-10 should be > 80% but in those tables, the corresponding number is around 70% [3].
4. The rationale for why using DPOT and its subsequent stealthiness across the defenses is unclear, given that Algo 1 and Algo 2 only focus on optimizing triggers to mislead the classification output of model $W_g$.
5. This method does not include any component to ensure the MA of the model, and the Backdoor attack should consider both ASR and MA. How to show that this method intuitively does not degrade MA. In Algo.2, there is no optimization objective to maintain clean accuracy, and the MA should be reported together with the ASR in the main paper.
6. The related works and introduction should include more works in the literature, now it is not extensive enough.

[1] Fang, Pei, and Jinghui Chen. "On the vulnerability of backdoor defenses for federated learning." Proceedings of the AAAI Conference on Artificial Intelligence. Vol. 37. No. 10. 2023.

[2] Nguyen, Thuy Dung, et al. "Iba: Towards irreversible backdoor attacks in federated learning." *Advances in Neural Information Processing Systems* 36 (2024).

[3] Xie, Chulin, et al. "Dba: Distributed backdoor attacks against federated learning." *International conference on learning representations*. 2019.

**Questions:**

1. In DPOT, the $W_g$ is used as the starting point model to optimize the trigger, however, at the original rounds, this model has not been converged, does it affect the performance of this method?
2. In Algo.1., why is not CE loss used here, given this is a classification problem?
3. How is the data poison rate set for the baselines? Since the high DPR may relate to higher suspicion, the original DPR in the DBA paper is much lower than 0.5. An experiment with a smaller DPR with all baselines can ensure fairness across baselines.
4. In Figure 3 and Table I, was it the non-IID or IID setting? It is important that the proposed method can work with both scenarios.

---

> ### Author Response · Authors · 2024-11-24
> **Response to Weakness 1**
>
> ***Weakness 1.1**: The trigger seems to be very noticeable, especially when the trigger size is high, which can not be effective against post-defense such as Neural Cleanse.*
>
> **Response**: Neural Cleanse is one of the most important defense strategies against backdoor attacks in centralized learning settings. FLIP extends Neural Cleanse’s post-defense strategy to federated learning (FL) settings, where benign clients perform trigger inversion and adversarial training using their local data to recover the global model from backdoors. We provided evaluation results and an extensive discussion of the DPOT attack against the FLIP defense in Appendix E.2, demonstrating that DPOT is effective against this post-defense strategy. Thank you for mentioning Neural Cleanse; we will add it to the references.
>
>
> ***Weakness 1.2**:The suggestion for the trigger size is vague and does not contain specific number ranges (Appendix D.3), which is hard to apply in real settings. Could the author argue more about it?*
>
> **Response:**
> Thank you for this suggestion! To quantify the visual stealthiness of a trigger, we use a computer vision model as the judge. In the following experiment, we chose a computer vision model over human vision not only because the model provides an objective assessment, but also because the benign performance of a clean model on poisoned data is significant to adversaries who aim to maintain the functionality of their own models while compromising a competitor's model.
>
> We trained a benign global model on clean data under the same training settings as the victim FL system, using it as the **judge model**. We consider a trigger to have good visual stealthiness if its poisoned data can maintain high benign accuracy on the judge model while showing a high attack success rate (ASR) on the victim global model. The following results explain why the DPOT trigger sizes differ across different datasets.
>
> **Judge model's performance on DPOT poisoned data**
> | | Trigger size| 0 | 25 | 64 | 100 |
> | --- | --- | --- | --- | --- | --- |
> |Fashion-| Benign acc |85.76 |79.32 |76.07 |70.53 |
> |-MNIST|Drop (%)| 0 | -7.5 | **-11.30**| -17.76 |
> |FEMNIST| Benign acc |81.24 |68.11 | 45.12 | 28.39 |
> |--- | Drop (%)|  0 | **-16.16** | -44.46 | -65.05 |
> |CIFAR10| Benign acc |70.81 | 52.98 | 35.90 | 25.06 |
> |--- | Drop (%)| 0 | **-25.18**| -49.30 | -64.61 |
> |Tiny- |Benign acc |43.44 | 42.32 | 35.89 | 29.53 |
> | -ImageNet| Drop (%)|  0 | -2.58 | **-17.38**| -32.02 |
>
> - Benign acc: accuracy of poisoned data being predicted to its original bengin label.
> - Drop (%): Benign acc drop compared to when testing clean data (trigger size is 0) on the Judge model.
>
> Based on the ASR results in Table 3 and Figure 3, we selected the trigger size by balancing the trade-off between attack performance and visual stealthiness—a larger trigger size results in a higher ASR but lower benign accuracy. We set the lower bound for 'Drop' at -30% and the lower bound for 'Final ASR' at 50%, and choose the smallest trigger size that meets both constraints.
>
> **Judge model's and Victim model's performance on poisoned data having different types of optimized triggers (CIFAR10)**
> | Attack |  None | DPOT | IBA | A3FL |
> | ---|---|---|---|---|
> | Benign acc | 70.81 | 52.98 | 7.98 |70.46  |
> | Drop (%) |0 | -25.18 |-88.73  | -0.49 |
> | Final ASR against Trimmed Mean | N/A | 100 | 10.2 | 35.0 |
>
> We compared the visual stealthiness and attack performance of different types of optimized triggers. We reproduced the triggers of IBA and A3FL using their default settings for CIFAR-10, as provided in their open-source projects. For $L_2$-norm constrained triggers like IBA's, the additive perturbations spread across the entire image cause a significant drop in the judge model's benign accuracy, indicating poor visual stealthiness to benign computer vision models. In contrast, for $L_0$-norm constrained triggers, like those used by IBA and DPOT, the judge model’s benign accuracy on poisoned data is only slightly reduced. Compared to A3FL's trigger, the DPOT trigger demonstrates better attack performance, as reflected in its higher Final ASR (Table 6).

---

> > ### Author Response · Authors · 2024-11-24
> > **Responses to Weakness 2 and Weakness 3**
> >
> > ***Weakness 2**: The authors did not well discuss the difference between this method with related works using optimized triggers nor comparing them [1][2]. The idea of optimizing triggers is somewhat similar to adversarial attack methods like the Basic Iterative Method (BIM) and Fast Gradient Sign Method (FGSM) in terms of iterative gradient-based optimization, but the discussion is missing.*
> >
> > **Response**: We cited and discussed the related work [1] in line 124 of the Related Work section. We would be eager to compare results with the method in [1] if their project becomes open-source in the future. The iterative gradient-based optimization method of the DPOT attack is inspired by adversarial attack methods like BIM and FGSM, but DPOT differs from them in its optimization objective (targeted vs. untargeted) and attack timing (before training vs. after training). We will include these methods in the Related Work section and add further discussion.
> >
> > Thank you for mentioning related work [2], which is a representative study using an $L_2$-norm constrained trigger to attack an FL system. We compared the attack results of IBA with our method, as shown in the table below.
> >
> > **Comparison results with IBA attack on CIFAR10**
> > | Measure | Final ASR| -------- | Average ASR| -------- | Main-task Acc |-------- |
> > | --- | --- | --- | --- | --- | --- | --- |
> > |  | IBA | Ours | IBA | Ours | IBA | Ours |
> > | FedAvg | 18.5| **100** | 26.0 |**98.5** | 69.3 |70.7 |
> > | Median | 21.8 | **100** | 14.2  |**96.1** | 69.8 |69.1 |
> > | Trimmed Mean | 10.2 | **100** | 12.2 |**88.6** | 69.7 |70.4 |
> > | RobustLR | 32.8 |**100**  | 33.7 |**98.6** | 70.3 |70.1 |
> > | RFA | 9.0 | **100** | 9.5 |**97.8** | 70.4 | 70.7|
> > | FLAIR | 0.1 | **62.3** | 3.7 |**50.7** | 70.5 |70.6 |
> > | FLAME | 3.9 |**59.8**  | 25.1 | **56.0**| 68.7 |70.3 |
> > | FoolsGold | 14.8 |**100**  | 13.4 |**98.5**| 70.1  |71.0 |
> > | FLCert | 4.0 | **99.2** | 4.2 |**88.3** | 69.3 |70.0 |
> > | Multi-Krum | 0.3 |**100**  |4.4  |**98.7** |64.3  |63.0 |
> >
> > In summary, our method demonstrates better attack effectiveness than IBA's method. To explain how we reproduced IBA's results, we integrated the entire attacker implementation from IBA's open-source project into our FL training framework and disabled any model-poisoning techniques to match our threat model. We faithfully used their attacker configuration for the CIFAR-10 dataset. All experiments were conducted under the same default settings listed in Table 5, with FL training configurations aligned with Appendix D.2.
> >
> >
> >
> > ***Weakness 3**: The results in Table 8 and Table 9 raise suspicion that in existing attack papers, the MA of CIFAR-10 should be > 80% but in those tables, the corresponding number is around 70% [3].*
> >
> > **Response**: According to the study in [4], the MA of an FL system can be influenced by many factors, including the non-IID setting, local training epochs, number of clients, local minibatch size, and others. Our FL training configurations, introduced in Appendix D.2, differ from those in [3], leading to a different MA. Existing studies on FL security have used diverse FL training settings, and as long as their settings align with the nature of FL, their results and conclusions remain valuable and recognized. For example, existing works [5-7] report a range of MA on CIFAR-10, from 39.57 to 92.6.
> >
> > [4] B. McMahan, E. Moore, D. Ramage, S. Hampson, and B. A. y Arcas. Communication-efficient learning of deep networks from decentralized data. In 20th International Conference on Artificial Intelligence and Statistics, pp. 1273–1282, 2017.
> >
> > [5] P. Riege, T. Krauß, M. Miettinen, A. Dmitrienko, and A. Sadeghi. Crowdguard: Federated backdoor detection in federated learning. In 31st ISOC Network and Distributed System Security Symposium, 2024.
> >
> > [6] K. Kumari, P. Rieger, H. Fereidooni, M. Jadliwala, and A. Sadeghi. Baybfed: bayesian backdoor defense for federated learning. In 44th IEEE Security and Privacy, 2023.
> >
> > [7] H. Zhang, J. Jia, J. Chen, L. Lin, and D. Wu. A3fl: Adversarially adaptive backdoor attacks to federated learning. In 36th Advances in Neural Information Processing Systems, 2024.

---

> ### Author Response · Authors · 2024-11-24
> **Responses to Weakness 4, Weakness 5, Weakness 6, and Question 1**
>
> ***Weakness 4**: The rationale for why using DPOT and its subsequent stealthiness across the defenses is unclear, given that Algo 1 and Algo 2 only focus on optimizing triggers to mislead the classification output of model.*
>
> **Response**: An intuitive explanation about the rationale is as follows:
> - In Algo1 and Algo2, we construct the backdoor objective by optimizing the backdoor trigger such that the current round’s global model exhibits minimal loss on the backdoored data.
> - When the global model becomes more optimized to the backdoored data, further training on this backdoored data will lead to smaller updates to the global model’s current state.
> - Therefore, when a malicious client’s local dataset is partially poisoned by the optimized trigger while the rest remains benign, the **model updates** produced on them can be **dominated by benign model updates** within a limited number of local training epochs, and thus the malicious client's model updates can conceal among benign ones.
> - According to study [8], malicious clients' model updates if constrained to have small difference among benign ones will lead to subsequent stealthiness across the defenses that focus on eliminating outliers.
>
> The theoretical justification that proves trigger optimization can cause small differences in a client’s model updates between the non-attacked and backdoored state can be found in Appendix C: Proposition C.1.
>
> Experiment results that help elaborate the working principles of DPOT attack can be found in section 5.2.2.
>
> [8] E. Bagdasaryan, A. Veit, Y. Hua, D. Estrin, and V. Shmatikov. How to backdoor federated learning. In 23rd International Conference on Artificial Intelligence and Statistics, pp. 2938–2948, 2020.
>
>
> ***Weakness 5**: This method does not include any component to ensure the MA of the model, and the Backdoor attack should consider both ASR and MA. How to show that this method intuitively does not degrade MA. In Algo.2, there is no optimization objective to maintain clean accuracy, and the MA should be reported together with the ASR in the main paper.*
>
> **Response**: Thanks for this suggestion. We would have liked to report our MA results alongside the ASR in the main paper, but since all backdoor attacks' MA results are similar or repetitive, we decided to move them to the appendix to save space for other results that provide more interesting observations.
>
> In addition to the experimental results in Tables 8, 9, and 10, which empirically demonstrate the DPOT attack's ability to maintain a normal MA, an intuitive explanation is that the poisoned data generated by the DPOT attack has minimal impact on a client's model updates after local training. As a result, the malicious client's model updates, when aggregated into the global model, have little effect on diverting the global model from its original benign update direction. This allows the global model to stay on track and consistently converge to a higher MA.
>
>
> ***Weakness 6**:The related works and introduction should include more works in the literature, now it is not extensive enough.*
>
> **Response**: Thank you for this suggestion. We found many valuable works and genuinely wanted to include them in the main body of the paper, but due to page limitations, we had to move some of them to the appendix. We would like to express our sincere respect and appreciation for all related works that help advance our community!
>
> ***Question 1**: In DPOT, the $W_g$ is used as the starting point model to optimize the trigger, however, at the original rounds, this model has not been converged, does it affect the performance of this method?*
>
> **Response**: Thanks for this insightful question! We added experiments to evaluate DPOT attack's extensive capability in attacking at different round and different frequency. The results are shown in the following table.
>
> **Final ASR of DPOT when attacking at different rounds and frequency on CIFAR10**
> | Start round | Interval round  |  FLAIR | FLAME |
> | --- |  --- |--- |--- |
> | 1 | 1 | 62.3| **59.8** |
> | 1 | 5 | 52.9| 53.8 |
> | 1 | 10 | 72.0 | 57.8 |
> | 50 | 1 | 52.1 | 44.9|
> | 50 | 10 | **76.5** | 48.5|
> | 100 | 1 | 54.2 | 50.3|
> | 100 | 10 |69.2 | 54.5|
>
> - Start round: The round that DPOT starts attacking.
> - Interval round: The number of rounds between two adjacent DPOT attacks.
>
> In conclusion, DPOT attack shows better attack effectiveness in lower attacking frequency when against specific defense strategy like FLAIR. FLAIR penalizes clients who are frequently flagged as suspicious by lowering their aggregation weight, reducing their impact on the global model. Malicious clients can regain their normal influence by pausing malicious behavior, allowing the penalty score to gradually decrease. For other defenses that do not use a similar strategy, the effectiveness of the DPOT attack shows insignificant variation across different start rounds and interval rounds.

---

> ### Author Response · Authors · 2024-11-24
> **Responses to Question 2 and Question 3**
>
> ***Question 2**: In Algo.1., why is not CE loss used here, given this is a classification problem?*
>
> **Response**: In the experiment implementation, we used CrossEntropy as the loss function (see Appendix D.2). We use MSE in Algorithm 1 as a loss example to maintain consistency with our theoretical analysis.
>
>
> ***Question 3**: How is the data poison rate set for the baselines? Since the high DPR may relate to higher suspicion, the original DPR in the DBA paper is much lower than 0.5. An experiment with a smaller DPR with all baselines can ensure fairness across baselines.*
>
> **Response**: Thank you for this suggestion. We implemented experiments to study the impact of different data poison rates on the performance of DPOT, FT, and DFT attacks. We evaluated Final ASR and Avg ASR to compare attack effectiveness, and Main-task Accuracy to assess the stealthiness of all backdoor attacks. Other attack and FL training settings are consistent with those in the main body of the paper. The results are shown in the following tables.
>
> ***Final ASR***
> | Data Poison Rate | 0.3 | --- | ---|  0.5 | --- | ---| 0.8 | --- | ---|
> |----|----|----|----|----|----|----|----|----|----|
> |   | Ours | FT | DFT | Ours | FT | DFT | Ours | FT | DFT |
> | Fedavg |**100.0**| **100.0**| 99.0| **100.0** |**100.0** |92.5 |**100.0**| **100.0**| 99.8|
> | Median |**100.0**| 97.0| 61.0| **100.0** |81.3 |72.0 |**100.0**| 96.3| 35.9|
> | Trimmed Mean |**99.7**| 97.3| 67.6| **100.0** |94.8 |38.3 |**99.9**| 95.5| 45.8|
> | RobustLR |**100.0**| **100.0**| 99.5| **100.0**|**100.0** |**100.0** |**100.0**| **100.0**| 99.7|
> | RFA |98.9| **100.0**| 94.0| **100.0** |**100.0** |98.2 |**100.0**| **100.0**| 79.3|
> | FLAIR |**54.9**| 17.6| 20.2|**62.3** |15.1 |9.7 |**40.1**| 14.2| 14.9|
> | FLCert |**99.4**| 96.7| 63.5| **99.2** |92.6 |33.7 |**99.9**| 89.1| 34.8|
> | FLAME |**99.9**| 26.6| 15.4| **59.8**| 19.0 | 16.8 |**24.3**| 12.1| 13.2|
> | FoolsGold |**100.0**| **100.0**| 98.8| **100.0** |**100.0** |94.1 |**100.0**| **100.0**| 98.3|
> | Multi-Krum |95.2| **100.0**| 77.3| **100.0**|**100.0** |99.9 |**100.0**| **100.0**| **100.0**|
>
> ***Average ASR***
> | Data Poison Rate | 0.3 | --- | ---|  0.5 | --- | ---| 0.8 | --- | ---|
> |----|----|----|----|----|----|----|----|----|----|
> |   | Ours | FT | DFT | Ours | FT | DFT | Ours | FT | DFT |
> | Fedavg |**97.3**| 92.5| 75.5| **98.5** |88.1 |50.2 |**98.3**| 93.1| 81.2|
> | Median |**93.4**| 68.2| 34.2| **96.1** |46.6 |41.8 |**96.4**| 59.2| 22.7|
> | Trimmed Mean |**88.0**| 68.5| 34.2| **88.6** |59.4 |22.6|**95.2**| 65.6| 29.6|
> | RobustLR |**97.9**| 94.7| 76.7|**98.6** |94.4 |87.4 |**98.2**| 92.2| 78.3|
> | RFA |**87.0**| 82.9| 53.5| **97.8** |81.0 |55.3 |**97.9**| 81.5| 45.7|
> | FLAIR |**50.1**| 16.2| 15.4|**50.7** |13.6 |10.1 |**45.3**| 14.1| 13.8|
> | FLCert |**93.8**| 65.7| 33.1| **88.3** |59.5 |21.3 |**91.4**| 61.8| 23.4|
> | FLAME |**84.3**| 51.6| 28.4| **56.0** | 18.2 | 14.4 |34.5| **43.1**| 40.4|
> | FoolsGold |**97.9**| 91.6| 72.8| **98.5** |88.4 |53.0 |**97.7**| 89.6| 71.9|
> | Multi-Krum |92.8| **98.3**| 56.1| **98.7**|**98.7** |82.8 |**99.6**| 98.7| 99.1|
>
> ***Main-task Accuracy***
> | Data Poison Rate | 0.3 | --- | ---|  0.5 | --- | ---| 0.8 | --- | ---|
> |----|----|----|----|----|----|----|----|----|----|
> |   | Ours | FT | DFT | Ours | FT | DFT | Ours | FT | DFT |
> | Fedavg | 71.2| 70.3| 70.2| 70.7 |70.4 |71.4 |69.3| 70.3| 70.1|
> | Median | 69.3| 69.8| 70.0| 69.1 |69.8 |69.7 |69.5| 69.0| 69.5|
> | Trimmed Mean |69.9| 69.8| 69.3| 70.4 |70.2 |70.8 |68.6| 68.9| 69.6|
> | RobustLR |70.4| 70.3| 70.7|70.4 |70.4 |70.4 |70.0| 70.7| 71.1|
> | RFA |70.4| 70.6| 70.5| 70.7 |70.3 |70.8 |70.6| 70.5| 70.0|
> | FLAIR |69.9| 69.2| 69.2|70.6 |71.0 |70.4 |69.8| 68.5| 69.7|
> | FLCert |70.7| 70.1| 68.9| 70.0 |69.8 |70.4 |69.1| 67.9| 69.9|
> | FLAME |69.3| 70.5| 70.7| 70.3 | 70.9 | 70.9 |68.6| 68.5| 69.3|
> | FoolsGold |70.5| 70.7| 71.4| 71.0 |71.2 |71.7 |70.1| 70.0| 70.4|
> | Multi-Krum |62.8| 60.1| 59.0| 62.9|63.2| 60.8|60.0| 61.1| 61.1|
>
> In general, DPOT attack shows better attack effectiveness than FT and DFT across different DPR. Compared to DPOT and FT attacks, we observed that DFT is truly more susceptible to DPR and is more effective with a small DPR.

---

> ### Author Response · Authors · 2024-11-24
> **Response to Question 4**
>
> ***Question 4**: In Figure 3 and Table I, was it the non-IID or IID setting? It is important that the proposed method can work with both scenarios.*
>
> **Response**: Thank you for this suggestion. All results in our paper are trained under non-IID settings, with the non-IID degree set to 0.5. We implemented experiments to study the impact of different non-IID degrees on the performance of DPOT, FT, and DFT attacks. We evaluated Final ASR and Avg ASR to compare attack effectiveness, and Main-task Accuracy to assess the stealthiness of all backdoor attacks. Other attack and FL training settings are consistent with those in the main body of the paper. The results are shown in the following tables.
>
> ***Final ASR***
> | Non-iid degree |  0 |--- | ---|  0.2 | --- | ---| 0.5 | --- |--- | 0.8 |--- | ---|
> |----|----|----|----|----|----|----|----|----|----|----|----|----|
> |   | Ours | FT | DFT | Ours | FT | DFT | Ours | FT | DFT | Ours | FT | DFT |
> | Fedavg |**100.0** |**100.0**| 99.7|**100.0** |**100.0**| 99.8| **100.0** |**100.0** |92.5 |**100.0** |**100.0**| 99.3|
> | Median |**100.0** |99.4| 72.2|**100.0** |99.0| 66.8|**100.0** |81.3 |72.0 |**99.3** |99.2| 33.0|
> | Trimmed Mean |**100.0** |99.4| 73.6|**100.0** |99.8| 86.8|**100.0** |94.8 |38.3 |**100.0** |96.0| 65.8|
> | RobustLR | **100.0**|**100.0**| 99.8|**100.0**|**100.0**| 99.8| **100.0**|**100.0** |**100.0** | **100.0**|**100.0**| 99.2|
> | RFA |**100.0** |**100.0**| 97.0|**100.0** |99.8| 95.5|**100.0**|**100.0** |98.2 |**100.0** |**100.0**| 98.5|
> | FLAIR |**49.7** |21.2| 12.9|**65.4** |29.3| 15.8| **62.3** |15.1 |9.7 |**19.9** |12.6| 5.1|
> | FLCert |**99.9** |99.7| 77.1|**100.0** |96.7| 54.1|**99.2** |92.6 |33.7 |**100.0** |92.1| 19.4|
> | FLAME |**46.5** |15.7| 15.1|**57.6** |16.7| 16.2|  **59.8**| 19.0 | 16.8 |**100.0** |8.5| 26.5|
> | FoolsGold |**100.0** |**100.0**| 99.9|**100.0** |**100.0**| 99.6|**100.0** |**100.0** |94.1 |**100.0** |**100.0**| 99.9|
> | Multi-Krum |**100.0** |**100.0**| 95.2|**100.0** |**100.0**| 18.9|**100.0**|**100.0** |99.9|**100.0** |**100.0**| **100.0**|
>
> ***Average ASR***
> | Non-iid degree |  0 |--- | ---|  0.2 | --- | ---| 0.5 | --- |--- | 0.8 |--- | ---|
> |----|----|----|----|----|----|----|----|----|----|----|----|----|
> |   | Ours | FT | DFT | Ours | FT | DFT | Ours | FT | DFT | Ours | FT | DFT |
> | Fedavg |**98.1** |94.6| 81.5|**97.9** |90.8| 80.9|**98.5** |88.1 |50.2 |**98.3** |94.2| 75.5|
> | Median |**95.9** |73.0| 39.2|**95.3** |79.5| 38.1|**96.1** |46.6 |41.8 |**89.0** |59.6| 19.6|
> | Trimmed Mean |**95.3** |75.6| 42.6|**96.7** |80.5| 53.7|**88.6** |59.4 |22.6 |**93.3** |49.4| 33.6|
> | RobustLR |**98.5** |95.0| 79.5| **98.6**|92.9| 81.7| **98.6** |94.4 |87.4 |**98.1** |95.1| 71.0|
> | RFA |**95.8** |86.6| 61.4|**97.5** |87.9| 61.5|**97.8** |81.0 |55.3 |**98.0** |85.8| 58.6|
> | FLAIR |**56.7** |27.7| 12.1|**64.9** |28.2| 14.5| **50.7** |13.6 |10.1 |**21.0** |9.3| 5.4|
> | FLCert |**96.7** |71.0| 40.4|**95.0** |74.7| 31.7|**88.3** |59.5 |21.3 |**96.2** |52.2| 13.7|
> | FLAME |**63.0** |38.8| 26.0|**56.1** |28.3| 21.4| **56.0** | 18.2 | 14.4 | **73.8**|21.1| 37.8|
> | FoolsGold |**98.4** |93.9| 81.4|**97.8** |92.6| 81.0|**98.5** |88.4 |53.0 |**98.7**|98.0| 80.0|
> | Multi-Krum |91.4 |**98.2**| 58.4|**98.6** |86.6| 15.1|**98.7**|**98.7** |82.8 |96.2 |**98.7**| 70.4|
>
> ***Main-task Accuracy***
> | Non-iid degree |  0 |--- | ---|  0.2 | --- | ---| 0.5 | --- |--- | 0.8 |--- | ---|
> |----|----|----|----|----|----|----|----|----|----|----|----|----|
> |   | Ours | FT | DFT | Ours | FT | DFT | Ours | FT | DFT | Ours | FT | DFT |
> | Fedavg |74.5 |74.5| 74.9|74.4 |74.3| 75.0|70.7 |70.4 |71.4 |55.9 |55.5| 56.2|
> | Median |74.5 |75.5| 74.3|73.8 |74.3| 74.9|69.1 |69.8 |69.7 |53.4 |52.9| 54.5|
> | Trimmed Mean |75.0 |74.3| 74.5|74.4 |75.2| 74.3|70.4 |70.2 |70.8 |52.7 |53.5| 54.5|
> | RobustLR |75.3 |75.1| 75.1|75.3 |74.4| 74.4| 70.4 |70.4 |70.4 |55.2 |55.8| 55.4|
> | RFA |74.3 |74.8| 75.1|75.0 |75.8| 74.4|70.7 |70.3 |70.8 |56.4|56.2| 55.2|
> | FLAIR |73.7 |73.4| 73.6|73.9 |72.7| 73.1| 70.6 |71.0 |70.4 |55.3 |52.5| 52.1|
> | FLCert |74.6 |74.0| 74.7|74.1 |74.8| 74.0|70.0 |69.8 |70.4 |53.9 |53.5| 53.3|
> | FLAME |73.2 |72.1| 73.0|72.9 |73.5| 73.5| 70.3 | 70.9 | 70.9 |56.1 |56.2| 57.5|
> | FoolsGold |74.1 |74.5| 74.5|74.6 |74.3| 74.5|71.0 |71.2 |71.7 |57.0 |55.1| 55.1|
> | Multi-Krum |73.8 |73.6| 73.1|73.4 |73.2| 73.0|62.9|63.2| 60.8|38.7 |41.0| 38.4|
>
> It can be observed from the last table that different non-IID degrees result in different main-task accuracies. A smaller non-IID degree indicates that the data distribution is closer to an IID distribution, with a non-IID degree of 0 representing an IID distribution. The DPOT attack consistently exhibits better attack effectiveness than FT and DFT, performing well across different non-IID degree settings.

---

> > ### Comment · Reviewer_qg3k · 2024-11-26
> >
> > Thanks to the authors for hard work and justification.
> > However, this work requires further refinement to clearly delineate its contributions compared to adversarial training methods such as BIM or FGSM, as well as related existing works in the literature. Additionally, the paper would benefit from deeper insights and a more comprehensive analysis, supported by new results, to enhance its value to readers.
> > A key concern remains the ability of the proposed method to balance ASR and MA effectively and the practicality of this method, both intuitively and theoretically. The reported MA values appear to have declined, which raises questions about the approach's overall efficacy.
> > Based on these considerations, I have increased my score to 5. However, I strongly recommend that the authors undertake an extensive revision to address the reviewers' comments and suggestions.

---

> > > ### Author Response · Authors · 2024-11-28
> > >
> > > Thank you for your constructive feedback and for increasing the score of our submission. Your thoughtful questions have helped us gain a better understanding of our work. We appreciate your suggestions and look forward to submitting a revised version that addresses these points.
> > >
> > > In response to the concern regarding the declined MA, we conducted additional experiments in an effort to improve it. We discovered that lowering the learning rate for local training helps the global model achieve a higher final MA. Specifically, by reducing the learning rate for the CIFAR-10 and ResNet18 settings to half of our original learning rate (0.025), we observed a final MA of 85.63, with the ASR remaining at a high level. We hope this result addresses the concern about our MA.

---

### Meta-Review · Area_Chair_GJnz · 2024-12-22

**Metareview:**

The authors propose a new method for backdooring federated learning called DPOT. The attack operates by optimizing the backdoor trigger to minimize its effect on model update, which achieves improved stealthiness and can bypass state-of-the-art defenses.

Reviewers raised several potential weaknesses, including the trigger being visually very noticeable, lack of discussion/comparison with prior work, and questionable threat model as the attack requires manipulation of specific pixel values. Reviewers were not able to reach consensus regarding the paper's final decision, and overall enthusiasm for acceptance remained low. Ultimately, AC believes while the paper makes notable technical contributions, its practical impact does not meet the standard of top-tier conferences such as ICLR.

**Additional Comments On Reviewer Discussion:**

Reviewers discussed the above weaknesses with the authors. In the rebuttal, the authors sufficiently addressed the concern of comparison with prior work, but were not able to sway the reviewers' opinion on the other weaknesses.

---

### Decision · Program_Chairs · 2025-01-22

Reject